# Control signal dimensionality depends on limb dynamics

Anna S. Korol[1], Valeriya Gritsenko[1,2]*

1 Department of Neuroscience, West Virginia University, Morgantown, West Virginia, United States of America, 2 Department of Human Performance, West Virginia University, Morgantown, West Virginia, United States of America

* vgritsenko@mix.wvu.edu

## Abstract

Neural control of movement has to overcome the problem of redundancy in the multidimensional musculoskeletal system. The problem can be solved by reducing the dimensionality of the control space of motor commands, i.e., through muscle synergies or motor primitives. Evidence for this solution exists; multiple studies have obtained muscle synergies using decomposition methods. These synergies vary across different workspaces and are present in both dominant and non-dominant limbs. We explore the effect of biomechanical constraints on the dimensionality of control space. We also test the generalizability of prior conclusions that muscle activity profiles can be explained by applied moments about the limb joints that compensate for dynamic and gravitational forces during reaching. These muscle moments derived from motion capture represent the combined actions of muscle contractions that are under the control of the nervous system. Here, we test the hypothesis that the control space dimensionality is shaped by the complexity of dynamic and gravitational forces. To achieve this, we examined muscle activity patterns across reaching movements in different directions, starting from different postures performed bilaterally by healthy individuals. We used principal component analysis to evaluate the contribution of individual muscles to producing muscle moments across different reaching directions and in both dominant and non-dominant limbs. Extending our earlier work, we find that muscle activity profiles are described well by muscle moment profiles during reaching by both dominant and non-dominant arms. Our results further show that the dimensionality of control signals depends on the complexity of muscle moments, supporting the primary hypothesis. Our results suggest that the neural control strategy for limb dynamics compensation involves the modulation of the co-contraction of proximal and distal antagonistic muscles that change limb stiffness.

**Data availability statement:** All relevant data are publicly available in the Figshare repository at the following DOI: https://doi.org/10.6084/m9.figshare.25360231.v3.

**Funding:** V.G. was supported by NIGMS grants P20GM109098 and P30GM103503. A.S.K. was supported by a fellowship from NIGMS T32 AG052375. This work was supported in part by the Office of the Assistant Secretary of Defense for Health Affairs through the Restoring Warfighters with Neuromusculoskeletal Injuries Research Program (RESTORE) under Award No. W81XWH-21-1-0138. Opinions, interpretations, conclusions, and recommendations are those of the author and are not necessarily endorsed by the Department of Defense. The funders had no role in study design, data collection and analysis, decision to publish, or preparation of the manuscript.

**Competing interests:** The authors have declared that no competing interests exist.

## Introduction

The production of movement involves integrating biomechanical, neural, and environmental factors. Muscle recruitment by the central nervous system (CNS) causes motion or lack thereof, such as when maintaining posture, through the nonlinear production of forces by muscles about the degrees of freedom (DOFs) of the joints. Body inertia, the number of DOFs, and external forces then shape the resulting posture and movement. This biomechanics is complex enough that neural sensorimotor circuits must embed its dynamics for efficient and robust control [1–7]. However, a problem of redundancy exists, i.e., the problem of choosing among multiple muscles and combinations of joint angles that are possible for a given desired hand position or motion. The plan for producing a goal-directed movement to visual targets is thought to originate in the premotor areas, where neural activity has many kinematic features in the extrinsic head-centered reference frame [8–10]. The kinematic redundancy problem may be resolved at the planning stage. The control signals to execute this movement originate in the primary motor cortex, where neural activity has many dynamic or force-related features in the intrinsic reference frame [11–15]. The transformations from motor plan into motor execution may potentially resolve the muscle redundancy problem through internal models. Internal models are neural circuits dedicated to embedding the anatomy and function of the musculoskeletal system [2,3,11,16], such as the muscle moment arms about joint DOFs [17], and the biomechanical relationships stemming from the inertial properties of the limb [18,19].

The solution to the redundancy problem can be observed through the reduced dimensionality of the neural control space, a concept known as muscle synergies or motor primitives. This concept implies that muscles (or motoneurons in the spinal cord) are recruited in groups that represent a specific action so that a combination of a smaller number of synergies can produce a larger number of different movements [20–24]. The anatomical organization of muscles that form agonistic and antagonistic groups that can be activated by fewer control signals constraining the dimensionality of control space [25]. The spatial organization of motoneuron pools in the spinal cord also captures the functional relationships of the muscles they innervate, further supporting the idea of reduced dimensionality of control space through neural embedding [16]. Indeed, the neural activation of muscles observed with surface electromyography (EMG) is reducible to a low-dimensional space for certain types of movements, including reaching [20,24,26]. However, the dimensionality reduction can be done using different mathematical methods applied to data organized in different ways [27,28]. These methods all obtain control space solutions with lower dimensions than the original dataset, but their neuromuscular or biomechanical underpinnings are not always clear.

Animal studies have worked out the basic modular and hierarchical organization of the neural motor control system that comprises nested feedback loops from sensory afferents to spinal motor neurons that encompass progressively more delayed processing circuits including the spinal reflexes, reticular formation in the brainstem, and primary sensory and motor cortices in the brain to name a few [29,30] (Fig 1A). Relating the synergies obtained with decomposition methods to the dynamic neural

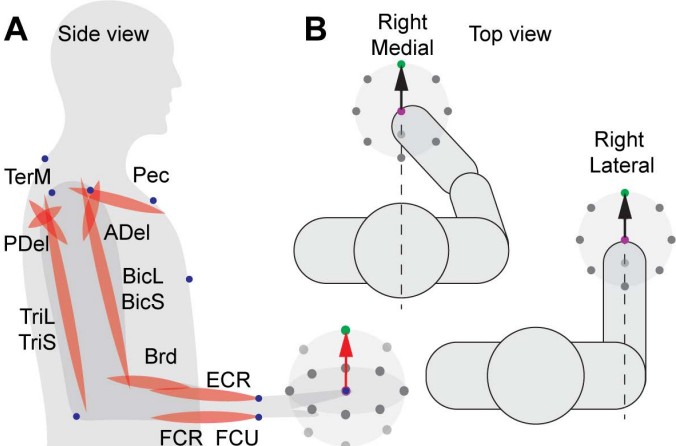

**Fig 1. Schematic of experimental setup. A.** Schematic of the anatomical arrangement of muscles recorded during the experiment. Abbreviations are as follows: the clavicular head of pectoralis (Pec), teres major (TerM), anterior deltoid (ADel), posterior deltoid (PDel), the long and lateral heads of triceps (TriL and TriS), the short and long heads of biceps (BiS and BiL), brachioradialis (Brd), flexor carpi radialis (FCR), flexor carpi ulnaris (FCU), and extensor carpi radialis (ECR). Blue dots indicate locations of LED markers for motion capture. Target locations illustrate the reaching directions from the central target (green). Drawing is not to scale. **B.** The target locations and the top view of the starting position in the medial and lateral workspaces for the right arm. The starting position is in magenta, the target – in green, and other targets – in shades of gray. The same reaching movements were also performed with the left arm in the left lateral workspace and the common central workspace.

control units is not trivial. The commonly used Principal Component Analysis (PCA) can be applied across muscles, experimental conditions, and time in two ways, to obtain temporally-variable synergies (temporal synergies) or temporally invariant synergies (spatial synergies) A recent study comparing spatial and temporal synergies has shown that these two types of synergies capture different features in EMG, which potentially reflect different aspects of neural control generating them [31]. Our earlier work shows that the biomechanical dynamics during reaching is captured accurately by the temporal synergies obtained with PCA. Specifically, the temporal dynamics of muscle torques, defined as rotational forces about joints, that act to support the arm against gravity and propel it toward the goal are well described by the 1st and 2nd temporal synergies, respectively, obtained with PCA [32]. Thus, the PCA method that obtains temporal synergies ensures that the forces needed to move the limb or limb dynamics are reflected in the signals with reduced dimensionality. Arguably, these signals are more likely to be represented at the neural circuit level because of the idea of the neural embedding of musculoskeletal dynamics, or internal models mentioned above, that enables the finite CNS circuitry to efficiently control movements in multiple directions from different starting postures at different speeds [33–35]. For example, the neural circuitry of the spinal central pattern generator that controls rhythmic behaviors, such as locomotion, by producing complex EMG patterns of multiple muscles spanning multiple joints is well understood [36–40]. A recent study has shown that the temporal dynamics of this spinal central pattern generator is captured well using temporal synergies obtained with PCA of EMG during locomotion [41]. This further proves that the temporal synergies obtained with PCA can accurately capture the known neural circuitry with complex temporal dynamics.

The temporal profile of the envelope of surface EMG is closely related to the force the corresponding muscle is producing in response to neural activation [42]. More recently, we have shown that for reaches with the dominant hand towards visual targets in three-dimensional space, the EMG profiles are closely related to the muscle torques that cause motion [32]. Specifically, the static component of EMG that underlies postural forces needed to support the arm in a specific posture and during transitions between postures is closely related to the components of muscle torques that include gravity terms in the equations of motion. This static component of EMG can be extracted as the 1st principal component of PCA applied to obtain temporal synergies [32]. This indicates that the static component of the EMG envelope from a

given muscle reflects the muscle's contribution to counteracting gravity load on the joints it spans. Moreover, the residual phasic component of EMG is closely related to the residual muscle torque that underlies the acceleration and deceleration forces toward the reaching goal after the gravity-related component is subtracted. This phasic component of EMG can be extracted as the 2nd principal component of PCA [32]. This indicates that the phasic component of the EMG envelope from a given muscle reflects the muscle's contribution to propulsion toward the target and stopping there. Here we will test the generalizability of these relationships between muscle activity and torque profiles to the non-dominant limb and across different workspaces.

Muscles are organized in a complex anatomical arrangement to generate synergistic and antagonistic or opposing torques around each joint DOF as determined by muscle moment arms [25]. The co-contraction of agonistic muscles and the reciprocal activation of antagonistic muscles directly contribute to the joint torques that are derivable from motion using classical mechanics. This pattern of muscle activity is typically observed during periodic movements controlled by the central pattern generator, such as walking, swimming, or cycling with legs or arms [43,44]. However, the concurrent activation of antagonists results in counterbalanced forces that cancel each other out in proportion to the corresponding muscle moment arms. These counterbalanced forces define the stiffness and viscosity components of impedance, which in turn determine the reaction of the musculoskeletal system to perturbations [45]. Both stiffness and viscosity components of mechanical impedance define how far the joints move in response to an applied external force [45]. The impedance parameters that are under the control of the nervous system can be measured by assessing changes in the co-contraction of antagonistic muscles [46]. This co-contraction increases during reaching movements with assistive inter-action torques [19,47] and in movements with unstable loads compared to those with stable loads [48]. The impedance of the whole limb increases to stabilize movement when learning novel dynamical tasks [49]. The joint stiffness component of impedance has been shown to be higher during unconstrained movements compared to constrained ones, indicating, unsurprisingly, that unconstrained movements are inherently less stable [50]. Limb impedance increases when learning to control an unstable object and when movement stability is challenged by a robot [51,52]. Wrist stiffness increases during unstable dynamical loads compared to stable ones [53] and when the accuracy demands of the task are high [54]. Therefore, dimensionality reduction of control space can be achieved through the co-contraction of both agonistic and antagonistic muscles. Here, we will examine the relative contribution of two strategies, 1) the control of direct joint torques with agonistic co-contraction and reciprocal activation and 2) the control of impedance through the antagonistic co-contraction, to the compensation for gravity and limb dynamics during unconstrained reaching.

In this study, healthy human participants performed unconstrained pointing movements toward visual targets located equidistantly from one of two central targets. The reaching movements in multiple directions were performed within the typical reaching workspace by either the dominant (right) or non-dominant (left) arm. We collected motion capture and EMG data and ran dynamic simulations with individualized inertial models of the arm to compute muscle torques and their components. We first tested the generalizability of the conclusions from Olesh et al. [32] by comparing principal component vectors obtained from muscle torques with those obtained from muscle activity (EMG profiles) across limbs and workspaces and quantified co-contraction using a novel method of analyzing PCA scores. PCA scores in our implementation represent the projection in the principal component space of each EMG profile during each reaching movement. The rationale is that reaching in different directions is caused by different amplitudes of postural or dynamic torques. Therefore, given that the EMG variance is captured well by these torques (generalization test above), the corresponding features in EMG also change with the amplitudes of the torques. For example, if the amplitude of shoulder flexion postural torque is larger during the movement up compared to the movement down, then the corresponding score for the 1st eigenvector for the anterior deltoid EMG (shoulder flexor) will also be larger during the movement up compared to the movement down. If the scores of multiple muscles change together across reaching directions, then the contributions of the eigenvectors to these muscles are also changing amplitudes together across reaching directions. This is evidence for not only co-contraction between muscles in a given movement but also across multiple movements in different

directions. This is also evidence that these co-contracting muscles can be activated by a single control signal represented by the eigenvector or the temporal synergy. When the muscles whose scores are changing together are agonists, this will capture agonistic co-contraction. Similarly, when the muscles whose scores are correlated are antagonists, this will capture antagonistic co-contraction that will change joint stiffness. When the scores are changing together, but they are of a different sign, this will capture reciprocal muscle activation that can also be driven by a single control signal through spinal reciprocal connections [37]. Overall, correlations between PCA scores will reveal the conditions when a single low-dimensional signal can recruit multiple muscles. In other words, the more muscles are co-activating, the fewer control signals are needed to recruit these muscles and the lower the dimensionality of control space is. Using this method, we tested the hypothesis that the dimensionality of control space is shaped by the complexity of dynamic and gravitational forces that needed to be compensated for by muscle contractions.

## Materials and methods

### Data collection

The experimental protocol was approved by the Institutional Review Board of West Virginia University (Protocol #1311129283). Perspective participants were recruited through fliers distributed around Morgantown, WV. The data collection started on March 12th, 2014, and ended on July 25th, 2016. Participants provided written informed consent prior to the start of experiments. A second member of the investigative team witnessed the signing of the consent form.

Nine healthy participants (mean age ± SD = 22.78 ± 0.67 years, 3 females and 6 males, sex assigned at birth) performed a modified center-out reaching task by pointing to visual targets in virtual reality (software Vizard by Worldviz, Oculus Rift) [32]. All participants reported they were right-handed and had no neurological or musculoskeletal disorders that could alter movement. Participants repeated reaching movements between a central target and one of 14 targets located along a sphere, with eight targets placed equidistantly on a horizontal circle parallel with the floor and eight targets placed equidistantly on a vertical circle parallel to the body sagittal plane, with two targets sharing locations in both circles as in Olesh *et al.* [32] (Fig 1). Participants were instructed to point to targets with the index finger while holding their hand palm down as quickly and accurately as possible without moving their trunk and wrist. The distances toward the targets were normalized for each subject based on their arm segment lengths to minimize the inter-subject variability in angular kinematics. Each movement to and from each target was repeated 15 times in a randomized order.

The outward and inward movements were repeated, with each arm starting at one of two locations of the central target. The central target was placed at the same distance relative to the participant's shoulder and scaled to their arm length. This further minimized the inter-subject variability in angular kinematics. For reaching within the lateral workspace, the central target was placed at a location that positioned the shoulder at 0 angle of all degrees of freedom (side view shown in Fig 1A) and the elbow at 90 degrees so that the upper arm was parallel to the trunk and the forearm with pronated wrist (hand facing down) was parallel to the floor (Fig 1B, bottom pictogram for top view). For reaching within the medial workspace, the central target was placed at a location that positioned the hand at the midline of the body and at the same distance from the trunk as in the lateral workspaces (Fig 1B, top pictogram for top view). The starting position was the same for reaches with both right and left arms, representing a common medial workspace. Thus, four conditions were created for reaches by the left and right arm in separate lateral workspaces and a common medial workspace.

Movement was recorded and visualized in virtual reality using the Impulse system (PhaseSpace Inc.) at a temporal resolution of 480 Hz. We tracked the location of 9 light-emitting diodes placed on the bony landmarks of the trunk, arm, and hand (Fig 1A) with 8 cameras, as described in detail in Olesh et al. [32]. EMG was recorded at a temporal resolution of 2,000 Hz using MA400-28 (MotionLab Systems). EMG was captured from 12 arm muscles: the clavicular head of pectoralis (Pec), teres major (TerM), anterior deltoid (ADel), posterior deltoid (PDel), the long and lateral heads of triceps (TriL and TriS), the short and long heads of biceps (BiS and BiL), brachioradialis (Brd), flexor carpi radialis (FCR), flexor carpi

ulnaris (FCU), and extensor carpi radialis (ECR). EMG recordings were temporally synchronized with the motion capture and virtual reality systems during reaching, as described in Talkington *et al.* [55].

## Data analysis

All data analysis was carried out in MATLAB 2023b (MathWorks Inc.). Motion capture data were low-pass filtered at 10 Hz and interpolated with a cubic spline. Arm kinematics was obtained from motion capture by defining local coordinate systems representing the trunk, arm, forearm, and hand. Joint angles representing 5 DOFs of the arm were derived using linear algebra, namely shoulder flexion-extension, shoulder abduction-adduction, shoulder internal-external rotation, elbow flexion-extension, and wrist flexion-extension (Fig 2A, example for one DOF). Active muscle torques and their components were calculated from kinematic data using a dynamic model of the arm with 5 DOFs (Simulink, MathWorks Inc.) as described in detail in [32]. Briefly, the individual height, weight, and segment lengths, together with the published anthropometric proportions, were used to scale the segment inertia in the model to individual body sizes [56]. Forearm, radioulnar, and hand segments were modeled as cylinders with lengths measured from individuals and the centers of mass defined by the anthropometric proportions. Joints were modeled as revolutes with the shoulder having 3 rotational DOFs, the elbow having 1 rotational DOF, and the wrist having 1 rotational DOF. The trunk was fixed in space and its inertia did not contribute to the joint torques. Inverse simulations were run with the angular kinematics as input and the active torques produced by muscles, termed muscle torques (MT), as output. Simulations were run with and without gravity force

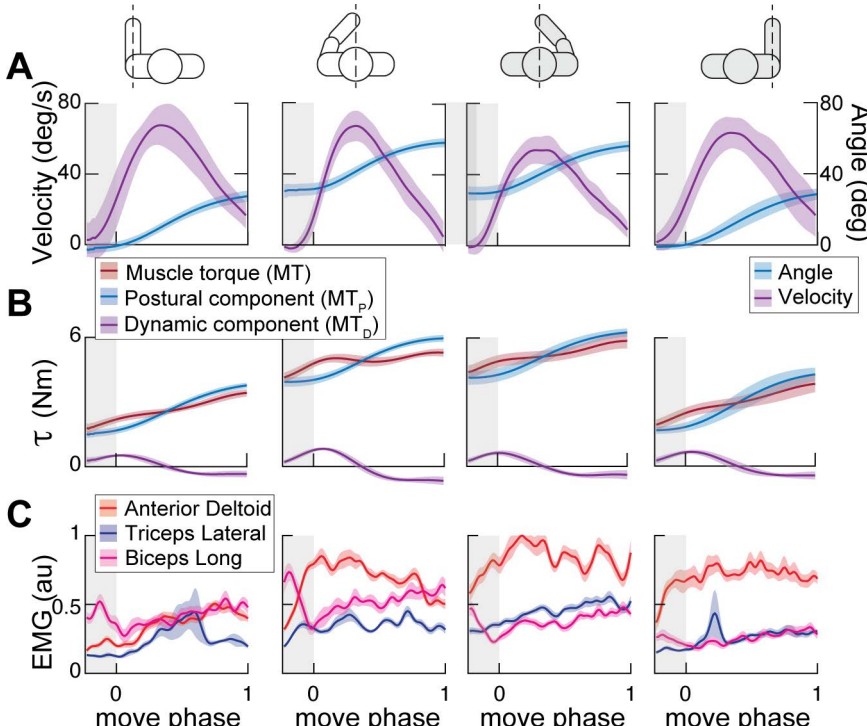

**Fig 2. Examples of kinematics, dynamics, and muscle activity profiles during reaching.** Each plot shows profiles per condition averaged across repetitions (n = 15) of the same reaching movement in one direction by one participant. **A.** Profiles (solid lines) and standard deviation (shaded areas) of shoulder flexion-extension angle and angular velocity. **B.** Shoulder flexion-extension torque (t) profiles that caused the movements in **A. C.** Normalized electromyography (EMG) profiles of anterior deltoid (ADel), triceps long (TriL), and biceps long (BiL) that accompanied the movements in **A.** Shaded areas show the standard error for the mean across repetitions (n = 15) of the same movement.

to estimate the components of muscle torque responsible for supporting the arm against gravity (the postural component of muscle torque or $MT_P$) and for intersegmental dynamics compensation (the dynamic component of muscle torque or $MT_D$) as described in detail in Olesh *et al.* [32]. The simulations were run for each trial. The numerical quality of inverse dynamic simulations was checked by running the same model in forward dynamics mode with gravity using the calculated muscle torques as inputs and simulated angular kinematics as outputs. The simulated and experimental joint kinematics were compared, and the mean ± standard deviation of the root-mean-squared differences between them was 0.05 ± 0.02 radians across all DOFs.

For subsequent analysis, torque profiles were normalized in time, averaged across the 15 repetitions per reaching direction per DOF and per subject, and down-sampled to 100 samples (Fig 2B, example for one DOF). Maximum torque amplitudes were determined per DOF per participant's arm and used to scale the amplitudes of averaged profiles across reaching directions and workspaces.

EMG recordings were low-pass filtered at 500 Hz, high-pass filtered at 20 Hz to remove motion artifacts, bandpass filtered at 59–61 Hz to remove electrical noise, rectified, and low-pass filtered at 10 Hz [57]. The last low-pass filter was applied to smooth the EMG envelope, which makes it more similar to the profile of force generated by the muscle [42]. Next, EMG recordings were normalized to movement duration. These durations were similar across the limbs and reaching workspaces (left lateral: 0.75 ± 0.10, left medial: 0.76 ± 0.06, right medial: 0.74 ± 0.08, and right lateral: 0.71 ± 0.08 seconds). Next, the profiles were down-sampled to 100 samples and averaged across the 15 repetitions per reaching direction per muscle. Maximum contraction values were determined for each muscle and participant across averaged EMG timeseries per reaching direction and used to scale the amplitudes of averaged profiles (Fig 2C).

Principal Component Analysis (PCA) was used to reduce the dimensionality of EMG and torque data, similar to Olesh *et al.* [32]. Normalized and demeaned EMG profiles from 12 muscles of 28 reaching movements (14 outward and 14 inward movements between each pair of targets) were combined into one matrix, ***An×m***, where n = 100 (time) and m = 336 (reaching directions x muscle). PCA was applied to the data for reaching with the right and left arm in either the lateral or medial workspace separately. Thus, four ***An×m*** matrices were obtained for each subject. Similarly, normalized and demeaned muscle torque profiles (MT, $MT_P$, and $MT_D$) from 5 DOFs of the 28 reaching movements were combined into one matrix, ***Bn×m***, where n = 100 (time) and m = 140 (reaching directions x DOF). Since the analysis included three types of torques (MT, $MT_P$, $MT_D$) in the four conditions, twelve ***Bn×m*** matrices were obtained for each subject. The temporal profiles of the 1st principal component (eigenvector) were aligned across subjects and conditions so that the signs of this component's scores carry the information about the positive or negative correlation with the signal. The 2nd eigenvector was not aligned; this was rectified with correlations and the carrying over of the negative sign to the scores of the corresponding subject or condition.

The assumptions for obtaining temporal synergies using PCA were that the biomechanical synergies that underlie joint torque production need to be the same across all the reaching directions (including outward and inward movements to and from each visual target) across all workspaces for both limbs. Several key assumptions for PCA were addressed [58]. The assumption of linear relationships is supported by prior studies showing that filtered and rectified surface EMG has a linear relationship with muscle force [42]. The assumption of large sample sizes is supported by the grouping of samples across multiple subjects, conditions, and signal types. The units of all variables (millivolts for EMG and newton meters for torques) are at an interval level. The variables are not perfectly collinear. Only the first two principal components with higher variance are retained, while those with lower variance are considered noise. The data was randomly sampled from the population to ensure that the results are generalizable. We have confirmed that the covariance matrix describes the distributions of EMGs and torques sufficiently well by using the Shapiro-Wilk test for normality of residuals (alpha = 0.05). We have also confirmed that the intrinsic dimensions are orthogonal by computing the dot product of each pair of eigenvectors from EMG and torques. The off-diagonal elements of the dot product matrix were approximately zero (less than $10^{-10}$), similar to an identity matrix. Thus, the principal components are orthogonal.

PCA was performed using *pca* function from MATLAB Statistics and Machine Learning Toolbox. Each matrix was centered, and a default singular value decomposition algorithm was applied to obtain eigenvectors (Fig 3), the variance accounted for (VAF) by each principal component (Table 1), and scores that are the representations of the matrix in the principal component space. It is important to note that reaching in the same direction leftward or rightward from the starting location relies on different muscles due to the bilateral symmetry of our body. That is why the reaches that include

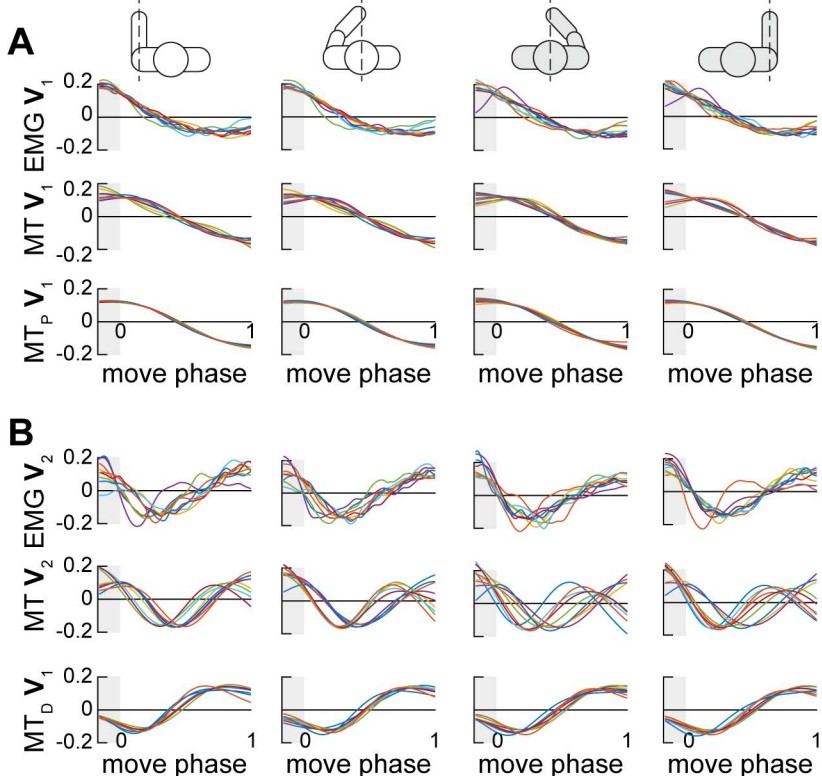

**Fig 3. Principal components of EMG and muscle torques.** Data from the four conditions indicated by the pictograms are shown in plots arranged in columns. Solid lines show profiles of features in time, colors indicate individuals. **A.** Gravity-related postural features in EMG and muscle torques. **B.** Propulsion-related dynamic features in EMG and muscle torques. EMG $V_1$ and EMG $V_2$ are the 1st and 2nd eigenvectors obtained from EMG, respectively; MT $V_1$ and MT $V_2$ are the 1st and 2nd eigenvectors obtained from muscle torques, respectively; $MT_P$ $V_1$ and $MT_D$ $V_1$ are the 1st eigenvectors obtained from the postural and dynamic components of muscle torque, respectively.

**Table 1. Variance accounted for during reaching with the right and left arm in the lateral and medial workspaces.**

|  | Left lateral | Left medial | Right medial | Right lateral |
|---|---|---|---|---|
| **EMG $V_1$** | 52 ± 8 | 49 ± 8 | 57 ± 6 | 54 ± 8 |
| **MT $V_1$** | 89 ± 5 | 88 ± 7 | 92 ± 4 | 91 ± 6 |
| **$MT_P$ $V_1$** | 98 ± 1 | 97 ± 4 | 97 ± 6 | 98 ± 1 |
| **EMG $V_2$** | 15 ± 4 | 15 ± 5 | 16 ± 5 | 17 ± 6 |
| **MT $V_2$** | 7 ± 3 | 9 ± 5 | 6 ± 3 | 7 ± 5 |
| **$MT_D$ $V_1$** | 79 ± 4 | 90 ± 4 | 93 ± 3 | 91 ± 7 |

Values are mean percentages ± standard deviation across 9 individuals. EMG – electromyography, MT – muscle torque. $MT_P$ and $MT_D$ are gravity-related postural and propulsion-related dynamic components of MT, respectively; $V_1$ and $V_2$ are eigenvectors.

leftward movements to the targets on the left from the central target performed by the left hand were compared to the mirror rightward movements to the right from the central target performed by the right hand and vice versa.

## Statistics and reproducibility

The first test of the generalizability of conclusions compared the profiles of the 1st and 2nd eigenvectors ($\mathbf{V}_1$ and $\mathbf{V}_2$) obtained from EMG to the eigenvectors obtained from muscle torques using correlation analysis (x*corr* function in MATLAB Signal Processing Toolbox). We then applied repeated measures analysis of variance (RM ANOVA, *ranova* function in MATLAB Statistics Toolbox) to test the null hypothesis that the coefficients of determination ($R^2$) from the four conditions and data types come from the same distribution. We used $R^2$ instead of the Pearson correlation coefficient because the former changes linearly and it is normally distributed. We verified that the normality of residuals was met using the Shapiro-Wilk test (custom function in MATLAB using *normcdf* and *norminv*). Equal sample sizes per condition per signal type were included in RM ANOVA. The multiple conditions included in ANOVA can have different variances, therefore, sphericity was tested using Mauchly's test (part of *ranova* function). If it is violated, the Greenhouse-Geisser correction will be used. In addition, we have tested for outliers using the z-score method (*zscore* function in MATLAB). Using a threshold of 3 standard deviations, no outliers were detected.

The between-subject factor in RM-ANOVA was Sex (assigned at birth), the within-subject factors were Conditions (2 workspaces for left and right arm each) and Components (comparing EMG $\mathbf{V}_1$ vs. muscle torque $\mathbf{V}_1$, EMG $\mathbf{V}_2$ vs. muscle torque $\mathbf{V}_2$, EMG $\mathbf{V}_1$ vs. postural torque $\mathbf{V}_1$, and EMG $\mathbf{V}_2$ vs. dynamic torque $\mathbf{V}_1$). Post-hoc multiple comparisons were done using the *multcompare* function in MATLAB Statistics Toolbox, which includes a correction for family-wise error rate using the Turkey's Honestly Significant Difference. The ethnicity–based statistical analysis was not performed due to the small sample size.

We have also analyzed statistically the amplitudes of PCA scores across different conditions. When the scores are close to 0, this means that the corresponding eigenvector does not capture much variance in a given EMG profile for a given reaching direction. For all combinations of muscles, movements, and participants, about 21% of the $\mathbf{V}_1$ scores were low, i.e., values were below 5% of the maximal score (24% for the right lateral workspace, 19% for the left lateral workspace, 17% for the left medial workspace, and 22% for the right medial workspace). About 26% of $\mathbf{V}_2$ scores were low (27% for the right lateral workspace, 26% for the left lateral workspace, 22% for the left medial workspace, and 28% for the right medial workspace). No linear relationships between the low score values would be expected across reaching directions because the low scores do not represent the EMG patterns with high fidelity. We have applied two RM ANOVAs (*ranova* function in MATLAB Statistics Toolbox) to $\mathbf{V}_1$ and $\mathbf{V}_2$ scores from all conditions using subject ID as the between-subject factor and the 4 conditions (right and left arm reaching in lateral and medial workspaces) as independent measurements. The Greenhouse-Geisser correction was used to correct for sphericity violation. The within-subject factors were designed to examine the changes in scores between outward and inward movements, between muscles (12 muscles, 66 comparisons), and reaching directions (only opposing targets were compared, e.g., scores for reaching leftward from the center vs. scores for reaching rightward from the center).

To test the hypothesis that the dimensionality of control space depends on the complexity of dynamic and gravitational forces, we evaluated how muscles co-activate across changing workspaces and reaching directions. The rationale is that reaching in different directions is caused by different amplitudes of postural or dynamic torques. Therefore, given that the EMG variance is captured well by these torques (1st test above), the corresponding features in EMG are also changing with the amplitudes of the torques. For example, if the amplitude of shoulder flexion postural torque was larger during movement upward compared to the movement forward, then the EMG $\mathbf{V}_1$ score of the anterior deltoid (AD, shoulder flexor) will also be larger during movement upward compared to the movement forward. If the scores of multiple muscles change together across reaching directions, then the contributions of the eigenvectors to these muscles also change amplitudes together across reaching directions (Fig 4). This is evidence for both co-contraction between muscles in a

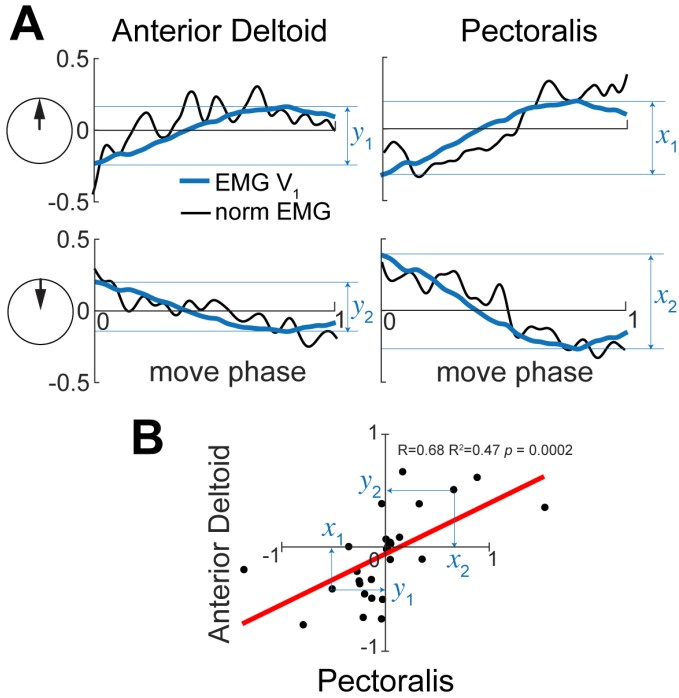

**Fig 4. Method for calculating linear regressions between PC scores across reaching directions. A.** Normalized electromyography profiles (norm EMG) and the first 2 principal components (EMG $V_1$ and EMG $V_2$) are shown for 4 reaching directions in the right lateral workspace by the same subject as in Fig 2. Arrows within circles illustrate the reaching directions as in Fig 1B. Blue labels show estimates of scores for EMG $V_1$ for two movement directions. **B.** Circles show scores for posterior deltoid (x axis) and anterior deltoid (y axis) across all movement directions, including outward and inward movements for the same subject. Scores labeled in **A** are shown with blue arrows. The line shows the linear regression, which was not significant in this instance.

given movement and across multiple movements in different directions. Moreover, these co-contracting muscles can be activated by a single control signal represented by the eigenvector or the temporal synergy. The advantage of this method of calculating the degree of co-contraction across multiple muscles is that it does not depend on the assumptions underlying the scaling of EMG amplitude. As long as the relative changes in the EMG amplitudes across reaching directions are preserved, the method works.

The method to use scores to measure co-contraction across multiple muscles and reaching workspaces was implemented using a linear correlation analysis (*corr* and *corrplot* functions in MATLAB Statistics and Econometrics Toolboxes). The correlations quantified the linear relationships across reaching directions between EMG $V_1$ or $V_2$ scores for each muscle pair combination. First, we extracted the $V_1$ scores that capture the gravity-related postural feature in EMG. We concatenated the scores into matrix $Cn×m$, where n = 28 (14 outward and 14 inward reaching directions) and m = 12 (muscles) per condition per subject. Outliers in score distributions for each muscle were removed using the *rmoutliers* function in MATLAB using the percentile range of 1.9 to 98.9%. A maximum of 12 score values were excluded from correlations per subject. The linear relationships were calculated between scores of 66 muscle pairs (Fig 4B). Positive correlations indicate co-contraction of muscles whose EMG profiles change together across reaching directions, while negative correlations indicate reciprocal activation of muscles, i.e., when the EMG activity increases in one muscle and decreases in another muscle. We evaluated the statistical significance of the linear relationships using the coefficient of determination ($R^2$). We applied Bonferroni-Sidak correction to correct the alpha for family-wise error so that the adjusted alpha = 0.0008. We repeated this analysis for the $V_2$ that captures the dynamic torque feature in EMG. Removing outliers did not change the results that were originally obtained with outliers.

## Results

Participants performed outward and inward movements with each arm, starting at one of two locations of the central target. This created four conditions for reaches by the left and right arm in separate lateral workspaces and a common medial workspace. All participants were able to complete the reaches with either left or right arms with consistent kinematics (Fig 2A). Due to bilateral symmetry, we expect that movements with left and right arms within the same workspace and kinematics in forward, backward, up, or down directions rely on similar forces produced by the same muscles. For the same reason, movements leftward or rightward by one limb will be similar to the rightward and leftward movements, respectively, by the other limb. The forces underlying these movements were inferred from motion capture using dynamic modeling that calculated muscle torques about each DOF of the shoulder, elbow, and wrist joints. These torques were further subdivided into a linear combination of postural and dynamic torques. The muscle torque profiles accompanying the corresponding movements had very similar trajectories across conditions (Fig 2B). However, there were static offsets in the postural torques underlying matching reaching directions. For example, holding the initial position at the beginning of the movement in both workspaces required larger muscle torques about the right shoulder flexion/extension DOF compared to that for the left shoulder (Fig 2B). This is likely due to the motor redundancy or the multiple combinations of joint angles and torques that can result in the same position of the hand.

From our earlier work, we knew that important features of muscle activity can be captured by the postural and dynamic torques [32]. However, in that study, we focused our analysis on the movements performed by the dominant arm in the lateral workspace. Here, the analysis encompasses different workspaces and both arms, which enabled us to evaluate the generalizability of this conclusion. We found that the values of intra- and inter-subject variability of EMG profiles were very low. The intra-subject variability or the average standard deviation of EMG profiles across reaching repetitions for muscles in the dominant right arm (normalized to maximal EMG of each muscle per subject) was 6% ± 4% (standard deviation across subjects, SD). The corresponding inter-subject variability of EMG of muscles in the dominant left arm was 6% ± 3%. The corresponding values of intra-subject and inter-subject variability of EMG in the non-dominant left arm were 7% ± 2% and 8% ± 5% respectively. This shows that the muscle activation profiles were somewhat more consistent across repetitions of the same movement type for the dominant arm compared to the non-dominant arm. Moreover, the EMG profiles also broadly reflected the differences in the joint torques across conditions (Fig 2C). In the example for the reach forwards and up in Fig 2, the amplitude of anterior deltoid activity increased in parallel with the increases in the postural component of the shoulder flexion muscle torque across conditions (Fig 2, $MT_P$ and ADel). Furthermore, the initial propulsive flexion torque in the dynamic muscle torque profile was preceded by a burst in the activation of the long head of the biceps (Fig 2, $MT_D$ and BiL, end of the burst is visualized). However, EMG profiles are notoriously noisy and difficult to interpret. Therefore, in the following analysis, we will broadly test the generalizability of conclusions from Olesh et al. [32] by focusing on the salient features in EMG obtained with principal component analysis (PCA; see details in Methods).

The 1st eigenvector ($V_1$) obtained from EMG matched the $V_1$ obtained from the muscle torques and postural torques (Fig 3A). The $V_1$ captures the gradually increasing or decreasing torque profiles associated with the changes in posture from the starting position to the final position at the reach target, reflecting the forces needed to support the arm against gravity. The temporal profile of the 2nd EMG eigenvector ($V_2$) matched the muscle torque $V_2$ and the dynamic torque $V_1$ (Fig 3B). This feature captures the acceleration/deceleration-related changes in torques associated with propulsion. The variance accounted for by the principal components from torques was very high (Table 1). This indicates a high degree of intralimb coordination. The variance accounted for by the EMG $V_1$ and $V_2$ was lower than that for the corresponding components from torques (Table 1), likely due to the higher variability of EMG profiles compared to torque profiles. Importantly, the first 2 principal components account for about 70% of the variance in EMG profiles. The non-linearities that are not captured by linear PCA are likely due to the complexities of motor unit recruitment order, activation-contraction coupling, and force-length and force-velocity muscle properties. Therefore, the following results only apply to the components

of muscle activity that we can relate to forces as captured by the principal components that are common between the force-related EMG envelopes and muscle torques.

In our earlier study of reaching from a single starting position by the right arm, we determined that EMG $V_1$ captures the static component of muscle recruitment necessary for postural support and EMG $V_2$ captures the phasic component of muscle recruitment [32]. We also showed that the profiles of these EMG components match the profiles of muscle torque components. Here we tested the generalizability of this conclusion to the non-dominant limb and to another reaching workspace. To test this, the degree of similarity between PCA features obtained from torques and EMG was examined using cross-correlation analysis (see *Methods*). The eigenvector profiles obtained from EMG and torques were highly correlated with shared variance above 50% in most subjects and in multiple conditions. Statistical analysis supports this statement. One-tailed repeated measures analysis of variance (RM ANOVA) on the coefficients of determination ($R^2$) between eigenvector showed a significant main effect ($F_{(15, 105)} = 9.93$, $p < 0.001$ with Greenhouse-Geisser correction) and an insignificant interaction between Sex and within-subject factors ($F_{(15, 105)} = 1.35$, $p = 0.274$). The EMG $V_1$ was highly correlated with muscle torque $V_1$ and postural torque $V_1$ during reaching within both workspaces by both arms; the EMG $V_2$ was also strongly correlated with muscle torque $V_2$ and the dynamic torque $V_1$ (Fig 5). This shows that the EMG $V_1$ captures the muscle recruitment necessary for generating forces to counteract gravity and the EMG $V_2$ captures the muscle recruitment necessary for generating forces for accelerating and decelerating the arm toward the target.

The similarity between the dynamic features (EMG $V_2$, muscle torque $V_2$, and dynamic torque $V_1$) was lower than that for postural features (EMG $V_1$, muscle torque $V_1$, and postural torque $V_1$). The shared variance captured by torque features in EMG was larger in males compared to females (mean squared error ± standard error of the mean = 0.10 ± 0.04, $p = 0.034$). There were differences in the similarities between features from different signals, with males showing higher similarity than females for the second principal component during right reaching in the lateral workspace (Tables 2 and 3). The differences between males and females are likely due to the larger size of males than females in our cohort (mean height of 6 males = 1.83 m, mean height of 3 females = 1.73 m; mean weight of males = 86.82 kg, mean weight of females = 60.94 kg).

The contribution of postural and dynamic features to EMG during movements in different directions varies with the changes in forces required to produce these movements. The EMG $V_1$ scores were significantly different from zero (main

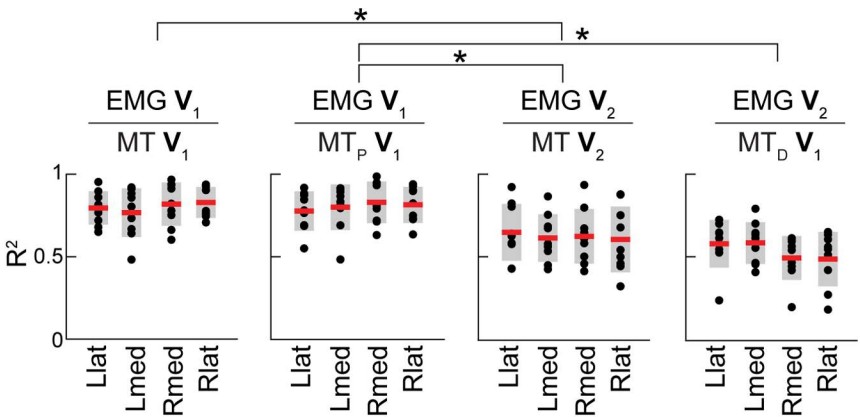

**Fig 5. Correlations between the temporal eigenvectors derived from EMG and muscle torques.** Dots show maximal $R^2$ values from individual cross-correlations between profiles in Fig 3, red lines show mean values, and grey boxes show standard deviations. EMG $V_1$ and EMG $V_2$ are the 1st and 2nd eigenvectors obtained from EMG, respectively; MT $V_1$ and MT $V_2$ are the 1st and 2nd eigenvectors obtained from muscle torques, respectively; $MT_P$ $V_1$ and $MT_D$ $V_1$ are the 1st eigenvectors obtained from the postural and dynamic components of muscle torque, respectively. Brackets and stars show significant differences summarized in Table 2.

**Table 2. Post-hoc differences in coefficient of determination between reaching workspaces and principal component features.**

| Workspaces | | MSE ± SE | p value |
|---|---|---|---|
| Left lateral | Left medial | 0.007 ± 0.023 | 0.990 |
| Left lateral | Right lateral | 0.031 ± 0.034 | 0.806 |
| Left lateral | Right medial | 0.017 ± 0.041 | 0.975 |
| Left medial | Right lateral | 0.024 ± 0.038 | 0.920 |
| Left medial | Right medial | 0.010 ± 0.037 | 0.993 |
| Right lateral | Right medial | -0.014 ± 0.028 | 0.957 |
| **Components** | | **MSE ± SE** | **p value** |
| EMG $V_1$/ $MT_P$ $V_1$ | EMG $V_1$/ MT $V_1$ | 0.009 ± 0.012 | 0.889 |
| EMG $V_1$/ $MT_P$ $V_1$ | EMG $V_2$/ $MT_D$ $V_1$ | 0.297 ± 0.057 | **0.005** |
| EMG $V_1$/ $MT_P$ $V_1$ | EMG $V_2$/ MT $V_2$ | 0.217 ± 0.044 | **0.007** |
| EMG $V_1$/ MT $V_1$ | EMG $V_2$/ $MT_D$ $V_1$ | 0.289 ± 0.050 | **0.003** |
| EMG $V_1$/ MT $V_1$ | EMG $V_2$/ MT $V_2$ | 0.208 ± 0.036 | **0.003** |
| EMG $V_2$/ $MT_D$ $V_1$ | EMG $V_2$/ MT $V_2$ | -0.081 ± 0.048 | 0.391 |

MSE – mean squared error, SE – standard error of the mean, EMG – electromyography, MT – muscle torque. $MT_P$ and $MT_D$ are gravity-related postural and propulsion-related dynamic components of MT, respectively; $V_1$ and $V_2$ are eigenvectors. Bold p values show significant differences with family-wise correction.

**Table 3. Post-hoc differences in coefficient of determination between males and females.**

| Workspaces | MSE ± SE | p value |
|---|---|---|
| left lateral M – F | 0.07 ± 0.06 | 0.320 |
| left medial M – F | 0.06 ± 0.07 | 0.405 |
| right medial M – F | 0.12 ± 0.05 | 0.056 |
| right lateral M – F | 0.16 ± 0.04 | **0.007** |
| **Components** | **MSE ± SE** | **p value** |
| EMG $V_1$/ postural τ $V_1$ M – F | 0.00 ± 0.07 | 0.987 |
| EMG $V_1$/ muscle τ $V_1$ M – F | 0.03 ± 0.06 | 0.564 |
| EMG $V_2$/ dynamic τ $V_1$ M – F | 0.20 ± 0.07 | **0.022** |
| EMG $V_2$/ muscle τ $V_2$ M – F | 0.16 ± 0.07 | **0.047** |

MSE – mean squared error, SE – standard error of the mean. Bold p values show significant differences with family-wise correction.

effect of RM ANOVA: $F_{(335, 9045)} = 23.99$, p < 0.000) and there was a significant interaction between subjects and within-subject factors ($F_{(335, 9045)} = 1.50$, p = 0.004). The EMG $V_2$ scores were not significantly different from zero ($F_{(335, 9045)} = 1.59$, p = 0.070), but there was a significant interaction between subjects and within-subject factors ($F_{(335, 9045)} = 1.47$, p = 0.003). The $V_1$ scores were larger for the outward movements compared to the inward movements (mean squared error ± standard error of the mean for outward – inward scores = 0.51 ± 0.03, p < 0.000) and, not surprisingly, larger than the EMG $V_2$ scores (Fig 6). This suggests that the neural control of muscle antigravity action during outward movements may be more stereotypical than during inward movements. In most cases, the outward and inward movements have an inverse change in EMG profiles, e.g., if the anterior deltoid increases its activity during the outward movement toward a given target to support the arm against an increasing gravitational load, then during the inward movement, the same EMG decreases for the same reason. Consequently, the sign of the score reflects whether a given EMG profile is positively or negatively correlated with the temporal profile of the eigenvector that captures this postural

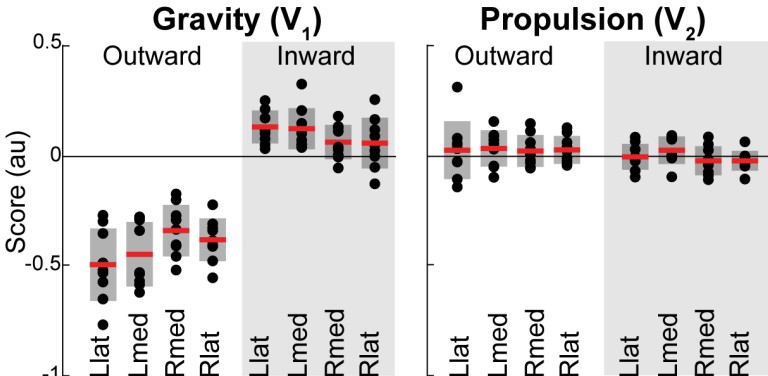

**Fig 6. Postural and dynamic scores from PCA on EMG for outward and inward reaches.** Dots show individual scores averaged across muscles and movement directions, red lines show mean values, and grey boxes show standard deviations. EMG $V_1$ and EMG $V_2$ are the 1st and 2nd eigenvectors obtained from EMG, respectively. Llat and Lmed indicate reaching with the left arm in lateral and medial workspaces, respectively; Rlat and Rmed indicate reaching with the right arm in lateral and medial workspaces, respectively.

feature. Indeed, the EMG $V_1$ scores across all participants switched signs between outward and inward reaches (Fig 7) in 68% of all reaching directions and muscles within the left lateral workspace, 70% and 68% within the medial workspace for the left and right arm, respectively, and 63% within the right lateral workspace. The EMG $V_2$ scores also switched signs in many instances (54% for left reaches in the lateral and medial workspace, 59% of samples during right reaches in the medial and lateral workspace). This supports the observation that the sign of the PCA scores in EMGs reflects the changing direction of the forces needed to reach different spatial locations.

Comparisons of scores between muscles have shown that the $V_1$ scores of most muscles spanning the shoulder were not different from each other (with the exception of scores of the posterior deltoid), while the $V_1$ scores for flexor carpi radialis were smaller than scores for most muscles that span the elbow and wrist (Table 4). There were also consistent differences in scores between some reaching directions (Table 5), further supporting the observation that the change in values of the PCA scores in EMGs reflects the changing amplitude of the forces needed to reach different spatial locations. There were no significant differences between $V_2$ scores across muscles or reaching directions (S1 and S2 Figs; S1 and S2 Tables), likely due to the low score values.

To test the hypothesis that the dimensionality of control space is shaped by the complexity of dynamic and gravitational forces, we examined the correlations between scores of PCA from EMG, as illustrated in Fig 4. The method takes advantage of the established relationships between torque and EMG components shown above and examines how the representations of the different muscles during different movements change in the principal component space. For example, in a given movement with large postural torques, we can expect higher $V_1$ scores for EMGs of one or more muscles involved in antigravity action compared to another movement with small postural torques and lower $V_1$ scores. If the EMG profiles of a pair of muscles change together across reaching directions to produce the antigravity action, the $V_1$ scores of these muscles should also change together across movements (Fig 4B). In this case, a significant linear relationship between the scores of the two muscles can be expected (e.g., Fig 9, ADel/Pec). Interestingly, the positive correlations indicate co-contraction in which both EMG profiles increase or decrease together in proportion to the temporal profile of the eigenvector vector (Fig 4). In contrast, a negative correlation indicates reciprocal activation in which one EMG profile increases while the other decreases in proportion to the temporal profile of the eigenvector. This indicates that the same neural control signal can activate the muscles whose scores are correlated across movements. Such activation would result in the combined muscle torque that is appropriate for supporting the arm against gravity in a given movement, representing a low-dimensional control signal for posture. A

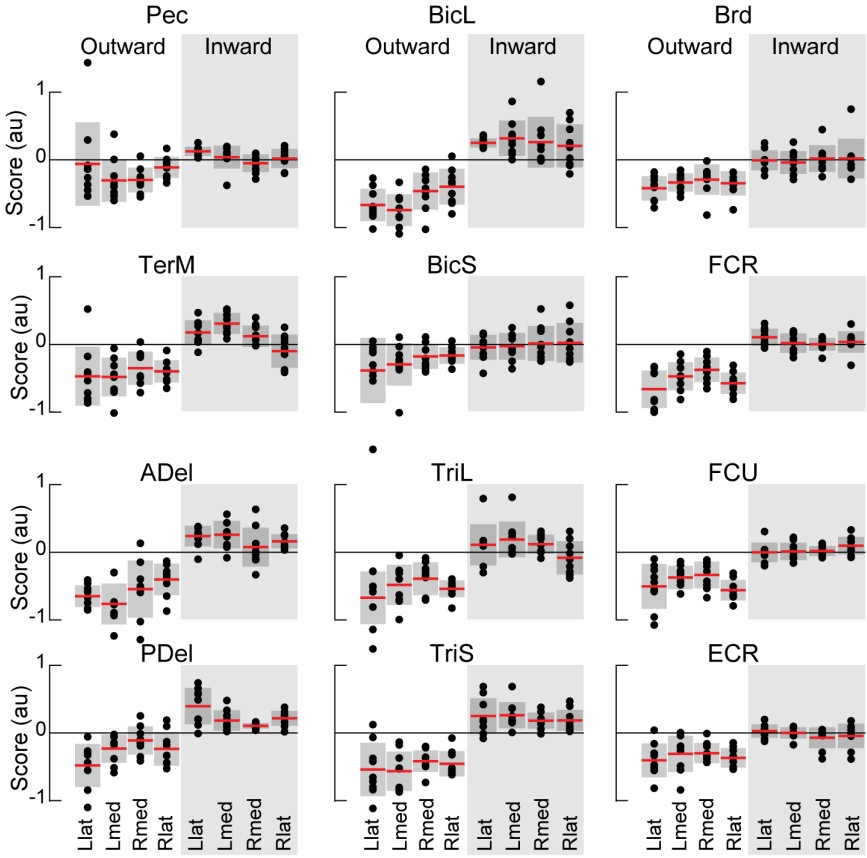

**Fig 7. Postural scores from PCA on EMG per muscle.** Dots show individual scores averaged across movement directions, red lines show mean values and grey boxes show standard deviations. EMG $V_1$ and EMG $V_2$ are the 1st and 2nd eigenvectors obtained from EMG, respectively. Llat and Lmed indicate reaching with left arm in lateral and medial workspaces, respectively; Rlat and Rmed indicate reaching with right arm in lateral and medial workspaces, respectively. Muscle abbreviations: the clavicular head of pectoralis (Pec), teres major (TerM), anterior deltoid (ADel), posterior deltoid (PDel), the long and lateral heads of triceps (TriL and TriS), the short and long heads of biceps (BiS and BiL), brachioradialis (Brd), flexor carpi radialis (FCR), flexor carpi ulnaris (FCU), and extensor carpi radialis (ECR).

similar logic applies to the $V_2$ scores (Example correlations in S3 Fig). Multiple significant linear correlations between scores of multiple pairs of muscles would indicate a broader co-activation of muscles that can be controlled by a common neural signal.

Multiple strong ($R^2 \geq 0.5$) linear correlations between $V_1$ scores of multiple pairs of muscles were observed across all participants (Fig 10). In contrast, very few strong linear correlations between $V_2$ scores were observed across all participants (Fig 11). More strong correlations between the $V_1$ scores were observed than between the $V_2$ scores (Fig 12). This was true for all individuals (S4–S12 Figs in S1 File). During reaching with the non-dominant left arm within the left lateral workspace, 32% of muscle pairs had their $V_1$ scores moderately or strongly correlated ($R^2 \geq 0.5$ in 21 pairs out of 66; Fig 10A). During reaching with the non-dominant arm within the medial workspace, 23% of muscle pairs had their scores moderately or strongly correlated ($R^2 \geq 0.5$ in 15/66 pairs; Fig 10B). In contrast, during reaching with the dominant arm, only 18% of muscle pairs had their $V_1$ scores moderately or strongly correlated during reaching within the shared medial workspace and 6% for the lateral workspace (Fig 10C and 10D, respectively). This shows that there appears to be a low-dimensional control strategy for generating the forces needed to support the arm against gravity. It is particularly prominent in the non-dominant arm (Fig 10A and 10B). In contrast, the few correlations between the $V_2$ scores suggest

**Table 4. Post-hoc differences in postural EMG V$_1$ score between muscles.**

| Muscle | MSE ± SE | p value |
|---|---|---|
| **PDel – TerM** | 0.1301 ± 0.0265 | **0.002** |
| **PDel – ADel** | 0.1837 ± 0.0165 | **0.000** |
| **PDel – BicL** | 0.1358 ± 0.0191 | **0.000** |
| **PDel – TriL** | 0.2004 ± 0.0318 | **0.000** |
| **PDel – TriS** | 0.1179 ± 0.0145 | **0.000** |
| **PDel – Brd** | 0.1570 ± 0.0198 | **0.000** |
| **PDel – ECR** | 0.1638 ± 0.0177 | **0.000** |
| **PDel – FCR** | 0.2203 ± 0.0203 | **0.000** |
| **PDel – FCU** | 0.1875 ± 0.0162 | **0.000** |
| **Pec – PDel** | -0.0622 ± 0.0268 | 0.487 |
| **Pec – TerM** | 0.0679 ± 0.0397 | 0.849 |
| **Pec – BicL** | 0.0737 ± 0.0371 | 0.699 |
| **Pec – BicS** | 0.0503 ± 0.0418 | 0.984 |
| **Pec – TriL** | 0.1382 ± 0.0469 | 0.180 |
| **Pec – TriS** | 0.0558 ± 0.0325 | 0.846 |
| **Pec – Brd** | 0.0948 ± 0.0379 | 0.380 |
| **Pec – ECR** | 0.1016 ± 0.0315 | 0.104 |
| **TerM – ADel** | 0.0537 ± 0.0316 | 0.855 |
| **TerM – BicL** | 0.0057 ± 0.0333 | 1.000 |
| **TerM – BicS** | -0.0177 ± 0.0403 | 1.000 |
| **TerM – TriL** | 0.0703 ± 0.0231 | 0.151 |
| **TerM – TriS** | -0.0122 ± 0.0257 | 1.000 |
| **TerM – Brd** | 0.0269 ± 0.0305 | 0.999 |
| **TerM – ECR** | 0.0337 ± 0.0240 | 0.953 |
| **TerM – FCR** | 0.0902 ± 0.0255 | 0.053 |
| **TerM – FCU** | 0.0574 ± 0.0241 | 0.448 |
| **ADel – Pec** | -0.1216 ± 0.0283 | **0.009** |
| **ADel – BicL** | -0.0479 ± 0.0169 | 0.222 |
| **ADel – BicS** | -0.0713 ± 0.0319 | 0.539 |
| **ADel – TriL** | 0.0166 ± 0.0339 | 1.000 |
| **ADel – TriS** | -0.0658 ± 0.0163 | **0.017** |
| **ADel – Brd** | -0.0268 ± 0.0208 | 0.974 |
| **ADel – ECR** | -0.0200 ± 0.0216 | 0.998 |
| **ADel – FCR** | 0.0366 ± 0.0224 | 0.883 |
| **ADel – FCU** | 0.0038 ± 0.0235 | 1.000 |
| **BicL – BicS** | -0.0234 ± 0.0279 | 0.999 |
| **BicL – TriL** | 0.0645 ± 0.0325 | 0.699 |
| **BicL – TriS** | -0.0179 ± 0.0142 | 0.978 |
| **BicL – Brd** | 0.0211 ± 0.0159 | 0.967 |
| **BicL – ECR** | 0.0279 ± 0.0235 | 0.986 |
| **BicL – FCU** | 0.0517 ± 0.0235 | 0.565 |
| **BicS – PDel** | -0.1124 ± 0.0317 | 0.052 |
| **BicS – TriL** | 0.0879 ± 0.0307 | 0.208 |
| **BicS – TriS** | 0.0055 ± 0.0324 | 1.000 |
| **BicS – Brd** | 0.0445 ± 0.0297 | 0.929 |
| **BicS – ECR** | 0.0513 ± 0.0336 | 0.919 |

*(Continued)*

**Table 4.** (Continued)

| Muscle | MSE ± SE | p value |
|---|---|---|
| BicS – FCR | 0.1079 ± 0.0311 | 0.062 |
| BicS – FCU | 0.0751 ± 0.0350 | 0.597 |
| TriL – TriS | -0.0824 ± 0.0296 | 0.242 |
| TriL – Brd | -0.0434 ± 0.0299 | 0.941 |
| TriL – ECR | -0.0366 ± 0.0286 | 0.975 |
| TriL – FCR | 0.0199 ± 0.0252 | 1.000 |
| TriL – FCU | -0.0129 ± 0.0296 | 1.000 |
| TriS – Brd | 0.0390 ± 0.0165 | 0.458 |
| TriS – ECR | 0.0458 ± 0.0164 | 0.236 |
| TriS – FCU | 0.0696 ± 0.0189 | **0.039** |
| Brd – ECR | 0.0068 ± 0.0181 | 1.000 |
| Brd – FCU | 0.0305 ± 0.0172 | 0.819 |
| FCR – Pec | -0.1582 ± 0.0376 | **0.011** |
| FCR – TriS | -0.1024 ± 0.0182 | **0.000** |
| FCR – BicL | -0.0845 ± 0.0218 | **0.024** |
| FCR – ECR | -0.0565 ± 0.0140 | **0.017** |
| FCR – Brd | -0.0634 ± 0.0130 | **0.002** |
| FCU – Pec | -0.1253 ± 0.0320 | **0.022** |
| FCU – FCR | 0.0328 ± 0.0152 | 0.589 |
| FCU – ECR | -0.0237 ± 0.0159 | 0.930 |

MSE – mean squared error, SE – standard error of the mean. Bold p values show significant differences with family-wise correction. Muscle abbreviations: the clavicular head of pectoralis (Pec), teres major (TerM), anterior deltoid (ADel), posterior deltoid (PDel), the long and lateral heads of triceps (TriL and TriS), the short and long heads of biceps (BiS and BiL), brachioradialis (Brd), flexor carpi radialis (FCR), flexor carpi ulnaris (FCU), and extensor carpi radialis (ECR).

that more complex and varied control signals are required for generating propulsive forces. Overall, these results support the hypothesis that limb dynamics shapes the dimensionality of control space solutions.

Correlations between scores were overwhelmingly positive, indicating that not only agonistic but also antagonistic muscles, such as biceps and triceps, changed their activity together. Note that in these reaching movements with the hand facing down, the recorded muscles performing the antigravity action are pectoralis, anterior deltoid, biceps, brachioradialis, and extensor carpi radialis (Pec, ADel, BicL, BicS, Brd, and ECR), while the rest produce muscle torques *in the* direction of gravity ([Fig 1]). Counterintuitively, most correlations were between the $V_1$ scores of antagonists, i.e., between scores in antigravity and "pro-gravity" muscles. Moreover, consistent correlations were observed between $V_2$ scores of antagonistic biceps and triceps muscles in both arms during reaching in both workspaces ([Fig 11]). This suggests that the low-dimensional control strategy for supporting the arm against gravity involves the modulation of limb stiffness through the co-contraction of antagonistic muscles.

The score coactivation patterns consisted largely of proximal and distal muscle groups. The proximal muscle group included the pectoralis, teres major, and deltoids. The distal muscle group included the brachioradialis, flexor and extensor carpi radialis, and flexor carpi ulnaris. Biceps and triceps muscles switched group allegiances most frequently between limbs, conditions, and individuals compared to other muscles ([Figs 10] and [11]). The scores had generally more correlations and were more strongly correlated within the group than between the groups in both arms ([Figs 10] and [11]). On average, when reaching within the lateral workplace, the correlations between the proximal shoulder muscle scores were

**Table 5. Post-hoc differences in postural V$_1$ score between reaching directions.**

| Target locations | MSE ± SE | p value |
|---|---|---|
| Horizontal plane,<br>left – right | 0.0039 ± 0.018 | 1.0000 |
| Horizontal plane,<br>toward – away from body | 0.0831 ± 0.0153 | **0.0007** |
| Horizontal plane,<br>back – forward | 0.0249 ± 0.0179 | 0.9750 |
| Horizontal plane,<br>lateral – medial | 0.0354 ± 0.0176 | 0.7484 |
| Vertical plane, down toward the body – up away from the body | 0.0392 ± 0.0177 | 0.6272 |
| Vertical plane,<br>down – up | -0.1328 ± 0.0203 | **0.0000** |
| Vertical plane, up toward the body – down away from the body | -0.0774 ± 0.0192 | **0.0222** |

MSE – mean squared error, SE – standard error of the mean. Bold *p* values show significant differences with family-wise correction. Target location pictograms and colors are as in Fig 8, compared reaching directions are combined.

more variable in the right arm compared to the left arm, while the correlations between the more distal muscles that span the elbow and wrist were more consistent across individuals in both arms (Fig 11A and 11C). Separate differences in correlations between either proximal or distal muscle scores were also observed in individuals (S4–S12 Figs in S1 File). This suggests that the recruitment of proximal and distal muscles during reaching may require separate control signals.

## Discussion

We observed that highly stereotypical muscle torques during reaching movements explain a high portion of the overall variance in EMG profiles in multiple arm muscles. We have shown that the first principal component of EMG captures the muscle recruitment necessary for generating forces to counteract the force of gravity, while the second principal component of EMG captures the muscle recruitment necessary for generating dynamic forces for accelerating and decelerating the arm toward the target (Fig 5). These conclusions, originally derived from a smaller subset of movements by the dominant arm in one area of the reaching workspace [32], are generalizable to movements by the non-dominant arm and across a larger reaching workspace that overlaps between both arms. This is useful for understanding the dimensionality of neural control space.

We have also observed correlations between scores of muscle pairs across reaching directions, indicating co-contraction primarily between antagonists, but which and how many muscles co-contracted varied under different conditions (Figs 10, 11 and S4–S12 Figs in S1 File). We have shown that the forces needed to support the arm against gravity can be generated by a few control signals, evidenced by the substantial number of co-contracting muscle pairs, particularly in the non-dominant arm. This is due to the force of gravity always pointing in the same direction, so that the

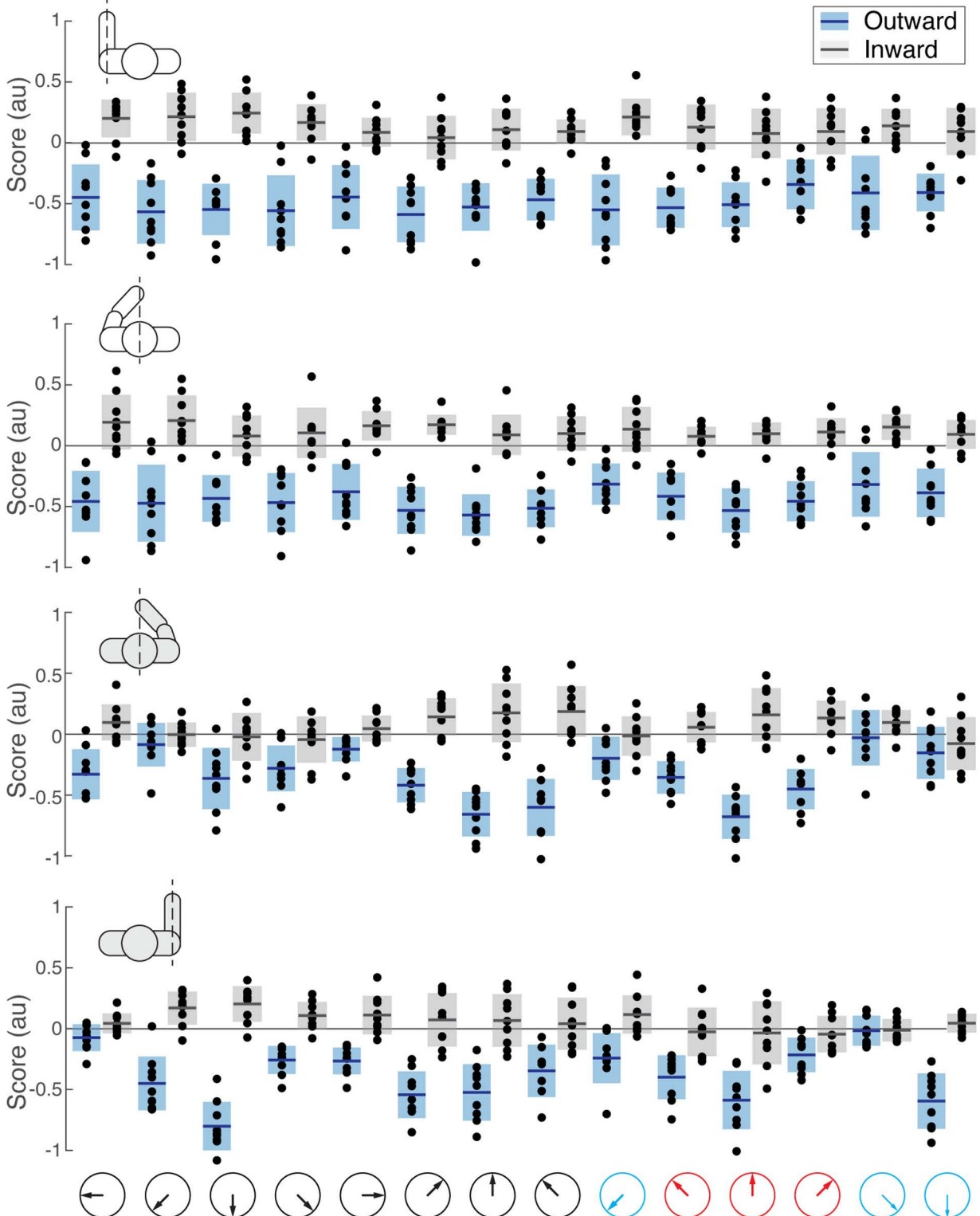

**Fig 8. Postural scores from PCA on EMG per reaching direction.** Data from the four conditions indicated by the pictograms are shown in plots arranged in rows. Dots show individual scores averaged across muscles, lines show mean values, and shaded boxes show standard deviations, blue indicates outward reaches, and grey indicates inward reaches. Circles with arrows show reaching directions; black circles indicate reaching in the

horizontal plane; blue circles indicate reaching downwards with gravity in the vertical plane; red circles indicate reaching upwards against gravity in the vertical plane.

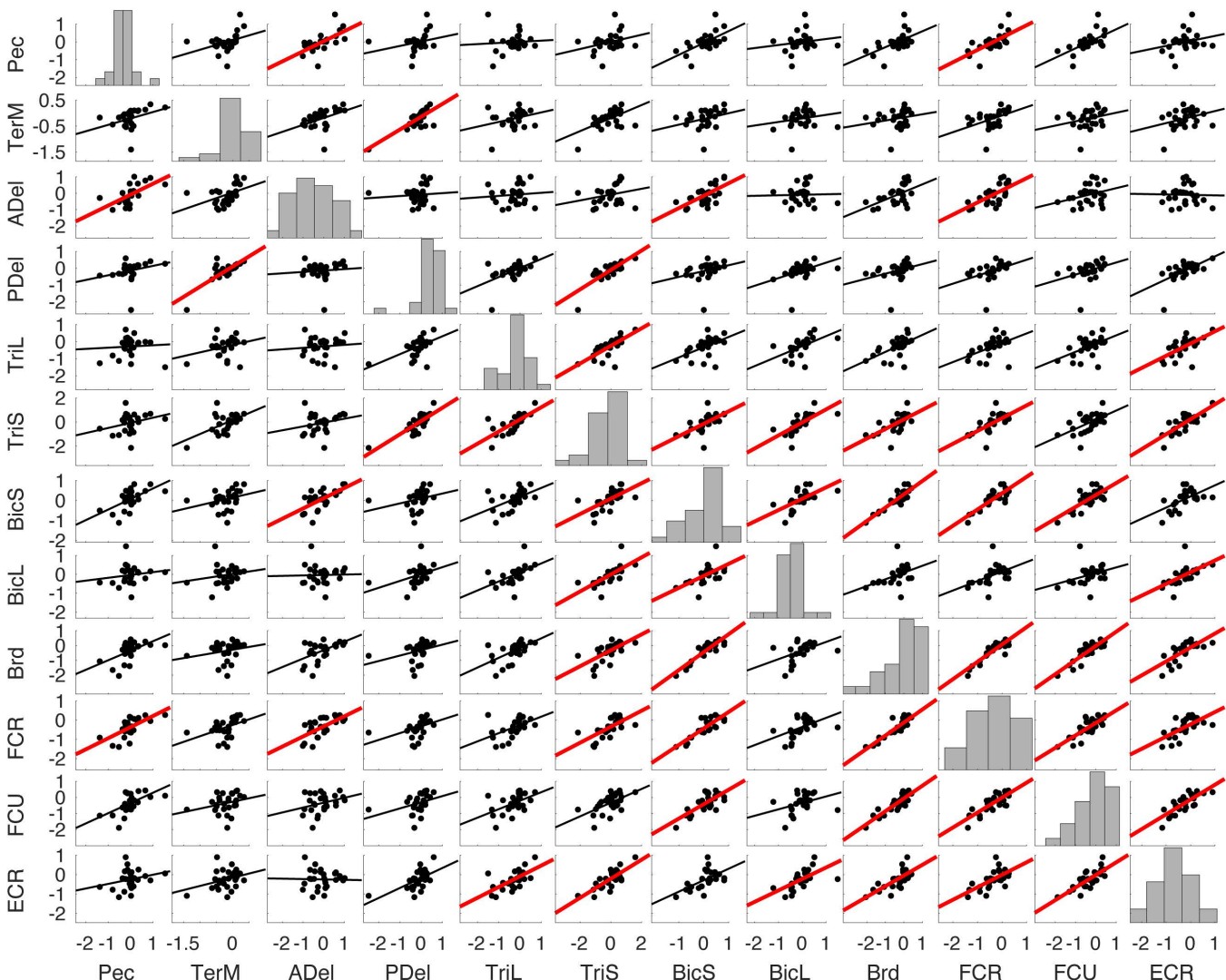

**Fig 9. Example regression matrix between postural scores from PCA on EMG.** The coordinates of each dot represent principal component scores for two muscles for a single reaching direction, as in Fig 4B. Muscle abbreviations: the clavicular head of pectoralis (Pec), teres major (TerM), anterior deltoid (ADel), posterior deltoid (PDel), the long and lateral heads of triceps (TriL and TriS), the short and long heads of biceps (BiS and BiL), brachioradialis (Brd), flexor carpi radialis (FCR), flexor carpi ulnaris (FCU), and extensor carpi radialis (ECR). Histograms along the diagonal show the distribution of the scores for a given muscle across reaching directions. Solid lines show least-squares linear regression, and red lines indicate significant relationships with correction for family-wise error, the adjusted alpha = 0.0008.

counteracting forces can be produced by consistent groups of muscles. We have also shown that the generation of the propulsive forces requires a higher dimensionality of control than that of the postural forces, evidenced by the lack of consistent co-contraction patterns across muscles. This is likely due to the direction-specific forces needed for reaching different targets that require different combinations of muscle forces. The exception to this rule was the consistent

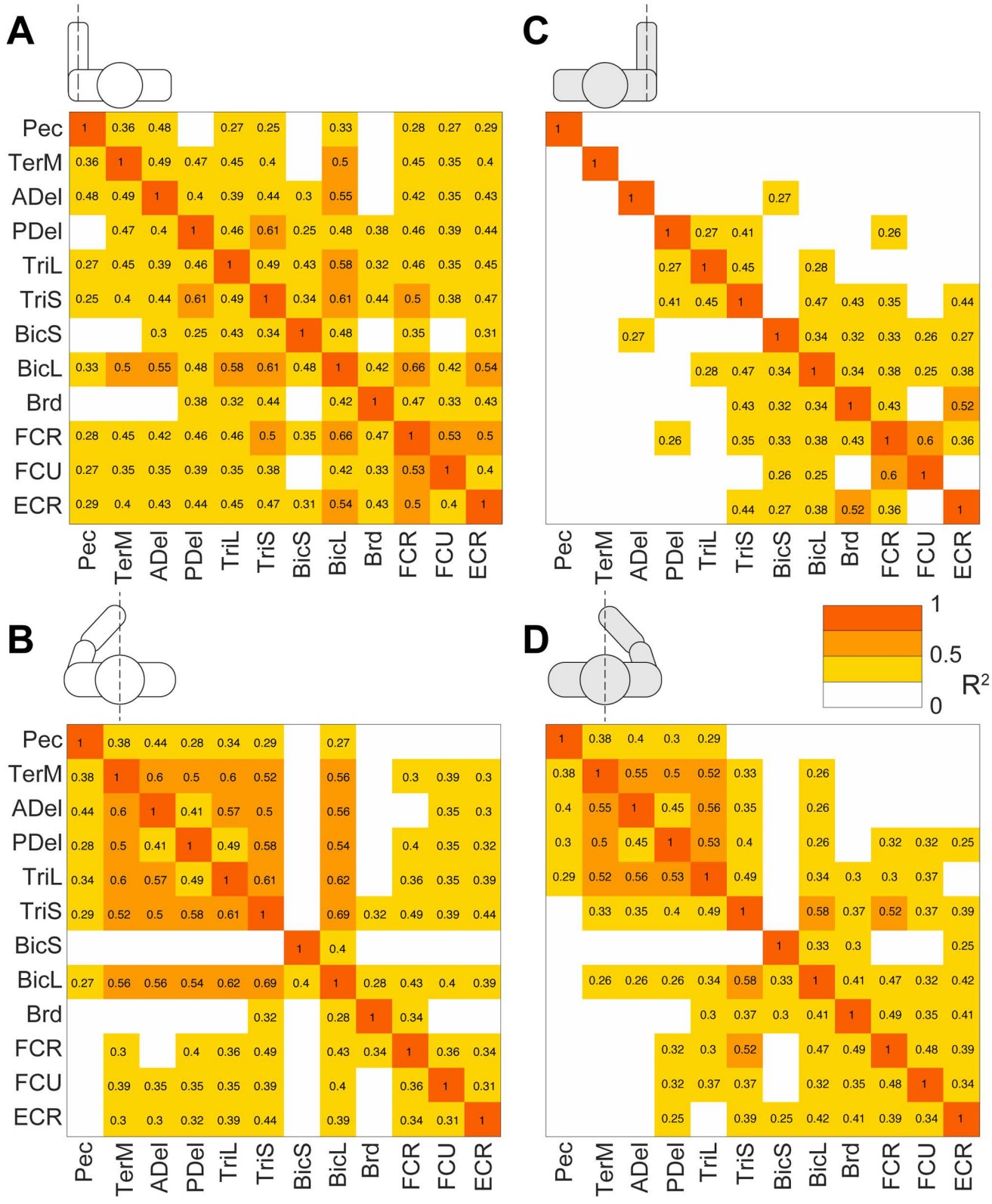

**Fig 10. Gravity-related muscle co-contraction.** Heatmaps show coefficients of determination ($R^2$) from regressions shown in Fig 10 averaged across 9 participants. Red and orange colors represent strong and moderate relationships, respectively, between EMG $V_1$ coefficients across reaching

directions. Pictograms indicate conditions for left reaching in lateral (A) or medial workspace (B) and for right reaching in lateral (C) and medial workspace **(D)**. Muscle abbreviations: the clavicular head of pectoralis (Pec), teres major (TerM), anterior deltoid (ADel), posterior deltoid (PDel), the long and lateral heads of triceps (TriL and TriS), the short and long heads of biceps (BiS and BiL), brachioradialis (Brd), flexor carpi radialis (FCR), flexor carpi ulnaris (FCU), and extensor carpi radialis (ECR).

co-contraction between the biceps and triceps muscles, whose function is likely to stabilize the elbow joint to compensate for interaction torques from the shoulder [18,19]. These results support our hypothesis that limb dynamics shapes the dimensionality of control space.

Our assumption in this work is that principal components represent a low-dimensional control signal that can be produced as a result of neural activity. Some evidence supporting this assumption comes from neural recordings in non-human primates [59]. The known organization of neural circuits with multiple nested feedback loops provides clues to how the different co-contraction patterns can be produced by a reduced set of control signals. For example, the co-contraction of agonists, such as between the elbow flexors biceps and brachioradialis, can be supported by common stretch reflex feedback from muscle spindles [60]. In this case, the co-contraction of agonists can be facilitated by a single descending signal that increases the gains of the stretch reflexes for these muscles. The negative reciprocal activation of antagonistic muscles, such as biceps and triceps, can be produced by inhibitory spinal interneurons, including Ia interneurons, that are known for their reciprocal inhibition action [61,62]. In this case, the reciprocal EMG can be facilitated by a single descending signal that excites the Ia interneurons for these muscles. The co-contraction of antagonistic action can be created through direct projections from the motor cortex, which is more extensive for motoneurons innervating distal muscles, through presynaptic inhibition of primary spindle afferents, the inhibition of Ia interneurons [63,64], or the spinal central pattern generator [37]. Although this analysis cannot distinguish between the contributions of different neural circuits to muscle recruitment, our method of analyzing the EMG co-contraction patterns can be used in future interventional studies, for example, with electrical or magnetic stimulation of specific neural structures, that can shed more light on the specific neural pathways that are involved in creating these patterns.

The reason we have not obtained consistent co-contraction patterns across movements may be that the surface EMG was too noisy and variable across the different movements and individuals. However, our variability data reported in the *Results* show that the temporal profiles of EMG were very consistent across all conditions, especially for the dominant arm with the fewest observed co-contractions. Another evidence of the consistency of EMG profiles is the substantial percentage of variance captured in EMG by the first two principal components (Fig 5) and by the components obtained from muscle torques (Fig 3), which are highly stereotypical during reaching movements. Therefore, it seems highly unlikely that the inter-movement or inter-subject variability can explain the lack of consistent co-contraction across different conditions.

Our results have shown that the component of EMG responsible for supporting the limb against gravity is present not only in antigravity muscles but also in their antagonists and that the amplitude of this component changes together across reaching directions (Fig 10). This is in line with other studies showing that limb impedance is an important control parameter during reaching movements [48,52,53,65]. Moreover, we observed more co-contraction in the non-dominant limb compared to that in the dominant limb, supporting earlier observations that there is hemispheric specialization in the compensation for limb dynamics. The non-dominant limb motor control is more specialized for impedance control [66–68]. Overall, this suggests that the neural strategy of controlling limb impedance helps to reduce the dimensionality of control space for complex muscle recruitment necessary for generating forces that support the limb against gravity during unconstrained reaching.

Our results also support the idea of internal models embedding body dynamics. Here, we have observed that a large amount of variance in the muscle activity profiles is captured by muscle torques. Interestingly, the component of muscle activity that reflects propulsion was not associated with consistent patterns of antagonistic co-contraction across multiple reaching directions (Fig 11). Instead, smaller subsets of consistent co-contraction were observed (S4–S12 Figs in S1 File). This provides evidence that control solutions with higher dimensionality are needed for generating the complex

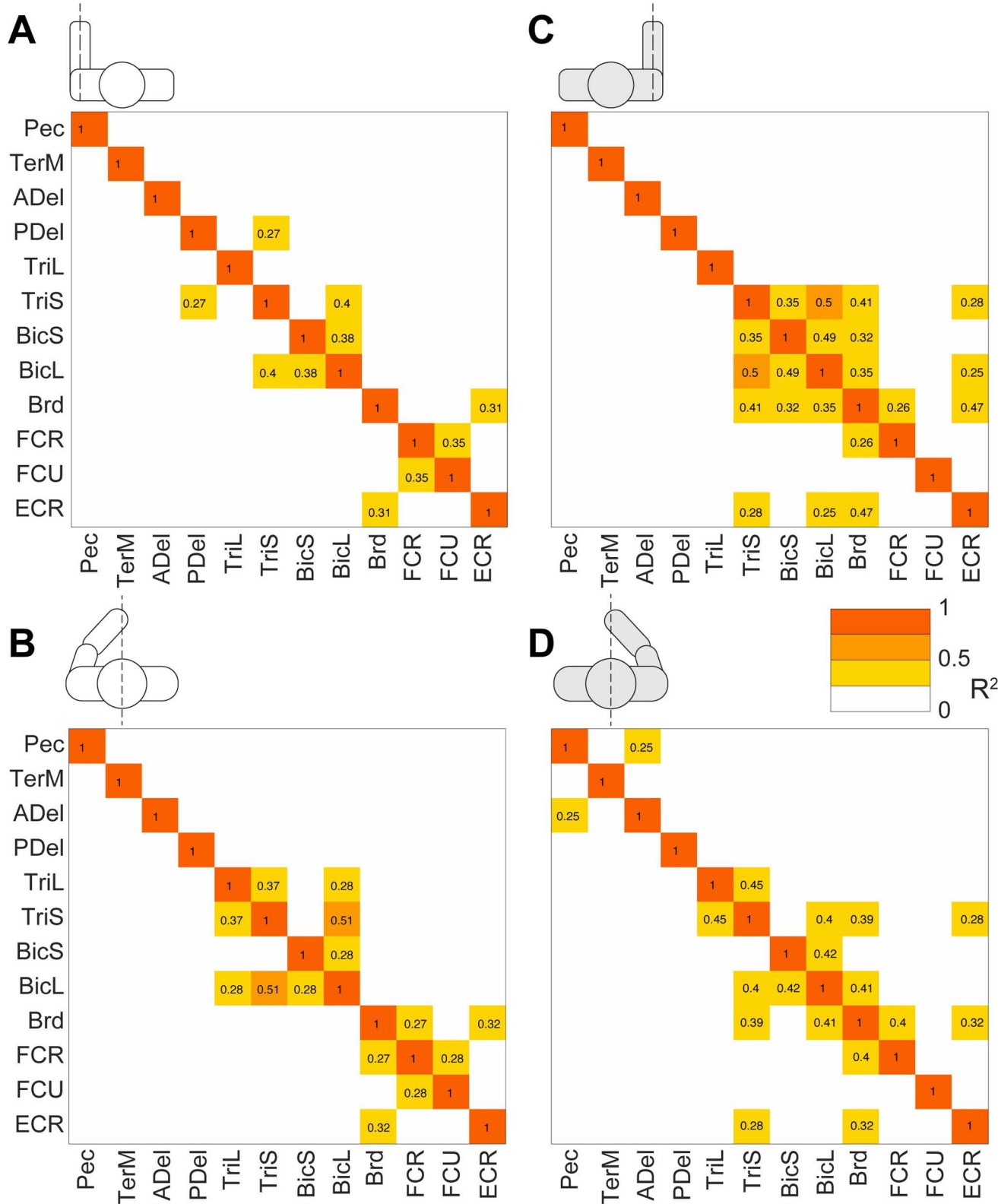

**Fig 11. Propulsion-related muscle co-contraction.** Heatmaps show coefficients of determination ($R^2$) from regressions shown in Fig 4 averaged across 9 participants. Red, orange and darker blue colors represent moderate and strong relationships between EMG $V_2$ coefficients across reaching

directions. Pictograms indicate conditions for left reaching in lateral (A) or medial workspace (B) and for right reaching in lateral (C) and medial workspace **(D)**. Muscle abbreviations: the clavicular head of pectoralis (Pec), teres major (TerM), anterior deltoid (ADel), posterior deltoid (PDel), the long and lateral heads of triceps (TriL and TriS), the short and long heads of biceps (BiS and BiL), brachioradialis (Brd), flexor carpi radialis (FCR), flexor carpi ulnaris (FCU), and extensor carpi radialis (ECR).

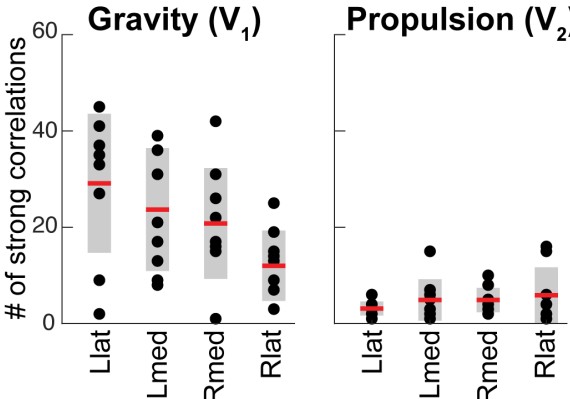

**Fig 12. The number of muscle pairs with correlated scores.** Dots show the number of moderate or strong (coefficient of determination $R^2 > 0.5$) linear regressions between pairs of different muscles out of 66 muscle pairs for each participant. Red lines and shaded areas show means and standard deviations, respectively. The left plot is based on regressions between EMG $V_1$ scores; the right plot is based on regressions between EMG $V_2$ scores. Llat and Lmed indicate reaching with the left arm in the lateral and medial workspaces, respectively; Rlat and Rmed indicate reaching with the right arm in the lateral and medial workspaces, respectively.

forces required to propel the limb in different directions and to stop at the goal. Internal models could support such high-dimensional control because the control solutions can be calculated on the fly by the neural networks that embed the equations of motion and biomechanical and anatomical constraints. These calculations likely involve the higher-order neural structures whose output converges on the corticospinal tract [19,69–71].

Our results further show that the recruitment of proximal and distal muscles during reaching varies independently. Separate differences in correlations between either proximal or distal muscle scores were observed in individuals and on average (S4–S12 Figs in S1 File, Figs 10 and 11). This supports our earlier results that the anatomical organization of muscles forms proximal and distal groups that can serve as separate control targets [25]. This mechanical organization is reflected in the spatial organization of motoneuron pools in the spinal cord that captures the functional relationships of the muscles they innervate [72]. Moreover, neural stimulation experiments have shown that cortical muscle representations are organized somatotopically based on their proximal-to-distal anatomy [73,74]. Distinct white matter tracts demonstrate some preferential contribution for proximally controlled gross movements versus distally controlled fine motor skills [75]. For example, responses to transcranial magnetic stimulation of the human motor cortex can facilitate differently the motoneuron pools of proximal and distal muscles depending upon the demands of a particular task, i.e., reaching vs. grasping [76]. The independent modulation of corticospinal excitability in proximal and distal muscles engaged in gravity compensation during reaching and posture maintenance has also been observed [77]. Altogether, our results provide supporting evidence for proximal and distal muscles being controlled independently during goal-directed movements [71].

## Conclusions

Results show that across multiple reaching directions, workspaces, and in both arms, muscle activity broadly reflects the temporal evolution of muscle torques. We suggest that the muscles generate forces to counteract gravity through

a simplified set of control signals, leading to coordinated contraction among multiple muscles of the whole limb, which adjusts the stiffness of the limb. Conversely, the forces that propel the limb demand more intricate control signals that coordinate separately proximal and distal muscles.

## Supporting information

**S1 Fig. Dynamic scores from PCA on EMG per muscle.** Dots show individual scores averaged across movement directions, red lines show mean values and grey boxes show standard deviations. EMG $V_1$ and EMG $V_2$ are the 1st and 2nd eigenvectors respectively obtained from EMG. Llat and Lmed indicate reaching with left arm in lateral and medial workspaces respectively; Rlat and Rmed indicate reaching with right arm in lateral and medial workspaces respectively. Muscle abbreviations: the clavicular head of pectoralis (Pec), teres major (TerM), anterior deltoid (ADel), posterior deltoid (PDel), the long and lateral heads of triceps (TriL and TriS), the short and long heads of biceps (BiS and BiL), brachioradialis (Brd), flexor carpi radialis (FCR), flexor carpi ulnaris (FCU), and extensor carpi radialis (ECR).
(TIF)

**S2 Fig. Dynamic scores from PCA on EMG per reaching direction.** Data from the four conditions indicated by the pictograms are shown in plots arranged in rows. Dots show individual scores averaged across muscles, lines show mean values, shaded boxes show standard deviations, blue indicates outward reaches, and grey indicates inward reaches. Circles with arrows show reaching directions; black circles indicate reaching in the horizontal plane; blue circles indicate reaching downwards with gravity in the vertical plane; red circles indicate reaching upwards against gravity in the vertical plane.
(TIF)

**S3 Fig. Example regression matrix between dynamic scores from PCA on EMG.** The coordinates of each dot represent principal component scores for two muscles for a single reaching direction, as in Fig 4B. Muscle abbreviations: the clavicular head of pectoralis (Pec), teres major (TerM), anterior deltoid (ADel), posterior deltoid (PDel), the long and lateral heads of triceps (TriL and TriS), the short and long heads of biceps (BiS and BiL), brachioradialis (Brd), flexor carpi radialis (FCR), flexor carpi ulnaris (FCU), and extensor carpi radialis (ECR). Histograms along the diagonal show the distribution of the scores for a given muscle across reaching directions. Solid lines show least-squares linear regression, and red lines indicate significant relationships with correction for family-wise error, the adjusted alpha = 0.0008.
(TIF)

**S1 File. S4–S12 Figs: Gravity and dynamic torque-related muscle co-contraction across all conditions for participants 1–9.** Heatmaps show coefficients of determination ($R^2$). Red, orange, and darker blue colors represent moderate and strong relationships between scores for the EMG $V_1$ (top row) and EMG $V_2$ (bottom row) across reaching directions. Pictograms indicate conditions for left reaching in the lateral or medial workspace and for right reaching in the lateral or medial workspace. Muscles are abbreviated as follows: the clavicular head of pectoralis (Pec), teres major (TerM), anterior deltoid (ADel), posterior deltoid (PDel), the long and lateral heads of triceps (TriL and TriS), the short and long heads of biceps (BiS and BiL), brachioradialis (Brd), flexor carpi radialis (FCR), flexor carpi ulnaris (FCU), and extensor carpi radialis (ECR).
(ZIP)

**S1 Table. Post-hoc differences in dynamic score between reaching directions.**
(PDF)

**S2 Table. Post-hoc differences in dynamic score between muscles.**
(PDF)

**S1 Checklist. PLOS One human subjects research checklist.**
(DOCX)

**S1 Text. Expedited approval letters 2014–2017.**
(PDF)

## Acknowledgments

We would like to acknowledge the contributions of Dr. E.V. Olesh and Dr. A.B. Thomas to the collection and preliminary analysis of the reported data.

## Author contributions

**Conceptualization:** Valeriya Gritsenko.

**Data curation:** Anna S. Korol, Valeriya Gritsenko.

**Formal analysis:** Anna S. Korol, Valeriya Gritsenko.

**Funding acquisition:** Valeriya Gritsenko.

**Investigation:** Anna S. Korol, Valeriya Gritsenko.

**Methodology:** Anna S. Korol, Valeriya Gritsenko.

**Project administration:** Valeriya Gritsenko.

**Resources:** Valeriya Gritsenko.

**Software:** Anna S. Korol, Valeriya Gritsenko.

**Supervision:** Valeriya Gritsenko.

**Validation:** Valeriya Gritsenko.

**Visualization:** Anna S. Korol, Valeriya Gritsenko.

**Writing – original draft:** Anna S. Korol, Valeriya Gritsenko.

**Writing – review & editing:** Anna S. Korol, Valeriya Gritsenko.

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
