## [Decision Letter · Decision Letter 0]

3 Sep 2024

PONE-D-24-24814Differential Impact of Biomechanical Constraints on Control Signal Dimensionality for Gravity Support Versus PropulsionPLOS ONE

Dear Dr. Gritsenko,

Thank you for submitting your manuscript to PLOS ONE. After careful consideration, we feel that it has merit but does not fully meet PLOS ONE’s publication criteria as it currently stands. Therefore, we invite you to submit a revised version of the manuscript that addresses the points raised during the review process. Please, revise the manuscript to address all questions raised by the reviewers regarding data analysis, statistics, and justifications of the conclusions. I recommend to follow reviewer's recommendations.  

We look forward to receiving your revised manuscript.

Kind regards,

Gennady S. Cymbalyuk, Ph.D.

Academic Editor

PLOS ONE

Journal Requirements:

"We would like to acknowledge the contributions of Dr. E.V. Olesh and Dr. A.B. Thomas to the collection and preliminary analysis of the reported data. V.G. was supported by NIGMS grants P20GM109098 and P30GM103503. ASK was supported by a fellowship from NIGMS T32 AG052375. This work was supported in part by the Office of the Assistant Secretary of Defense for Health Affairs through the Restoring Warfighters with Neuromusculoskeletal Injuries Research Program (RESTORE) under Award No. W81XWH-21-1-0138. Opinions, interpretations, conclusions, and recommendations are those of the author and are not necessarily endorsed by the Department of Defense."

"V.G. was supported by NIGMS grants P20GM109098 and P30GM103503. ASK was supported by a fellowship from NIGMS T32 AG052375. This work was supported in part by the Office of the Assistant Secretary of Defense for Health Affairs through the Restoring Warfighters with Neuromusculoskeletal Injuries Research Program (RESTORE) under Award No. W81XWH-21-1-0138."

"V.G. was supported by NIGMS grants P20GM109098 and P30GM103503. ASK was supported by a fellowship from NIGMS T32 AG052375. This work was supported in part by the Office of the Assistant Secretary of Defense for Health Affairs through the Restoring Warfighters with Neuromusculoskeletal Injuries Research Program (RESTORE) under Award No. W81XWH-21-1-0138."  

4. Please note that your Data Availability Statement is currently missing a direct link to access each database. If your manuscript is accepted for publication, you will be asked to provide these details on a very short timeline. We therefore suggest that you provide this information now, though we will not hold up the peer review process if you are unable.

5. Thank you for uploading your study's underlying data set. Unfortunately, the repository you have noted in your Data Availability statement does not qualify as an acceptable data repository according to PLOS's standards.

Reviewers' comments:

Reviewer's Responses to Questions

**Comments to the Author**

1. Is the manuscript technically sound, and do the data support the conclusions?

Reviewer #1: Yes

Reviewer #2: Partly

2. Has the statistical analysis been performed appropriately and rigorously? 

Reviewer #1: Yes

Reviewer #2: No

3. Have the authors made all data underlying the findings in their manuscript fully available?

Reviewer #1: Yes

Reviewer #2: No

4. Is the manuscript presented in an intelligible fashion and written in standard English?

Reviewer #1: Yes

Reviewer #2: No

5. Review Comments to the Author

Reviewer #1: In this paper authors explore the effect of biomechanical constraints on the dimensionality of control space. They showed that the muscle torques that support the limb against gravity are produced by more consistent combinations of muscle co-contraction than those that produce propulsion.

It is not clear to me why authors did PCA analysis of combined from different motor tasks data. If it were one motor task, it would show that CNS reduces control space to control this particular motor task. But what is possibly could be controlled for all different types of motions combined? Each motor task should be controlled differently. I would recommend to re-do analysis for each motor task separately.

The authors wrote that there were 15 repetitions of each movement but in matrix A there is one EMG per motion. I suggest that analysis was done for each repetition separately, but it was not mentioned in the paper.

The left part of Figure 1 is not necessary. Instead, it would be better to show locations of targets for different conditions, at least for one side.

Figure 2 has angles and angular velocities plotted on one panel. I would recommend using left/right different vertical axis as well using degrees instead of radians for angles.

pp.55-56 Authors wrote: “target was placed at a location that positioned the shoulder at 0 angle of all degrees of freedom and elbow at 90 degrees”

For some angles ‘0’ could be set in different positions. Was wrist pronated or supinated? Was wrist also parallel to the floor?

pp.82-85 Authors wrote: “Joint angles representing 5 DOFs of the arm were derived using linear algebra, namely shoulder flexion-extension, shoulder abduction-adduction, shoulder internal-external rotation, elbow flexion-extension, forearm pronation-supination, and wrist flexion-extension”

Are there 5 or 6 DOFs?

shoulder flexion-extension(1), shoulder abduction-adduction(2), shoulder internal-external rotation(3), elbow flexion-extension(4), forearm pronation-supination(5), and wrist flexion-extension(6)

Reviewer #2: Korol et al. examined muscle activity patterns during reaching movements in different directions and postures, performed by healthy individuals. They tested the relationship between muscle torques and muscle activity profiles, and the influence of biomechanical constraints on the dimensionality of control space. Using principal component analysis, they evaluated the contribution of individual muscles to joint torques. This work is an extension of a previous paper, the main novelty is that both dominant and non-dominant arm movements were included. They conclude that muscle torques supporting the limb against gravity are produced by more consistent combinations of muscle co-contraction than those producing propulsion. There are several issues in the manuscript that need clarification. I list them in the following.

Major:

1. The paper is quite difficult to follow. This is mainly due to the writing style, use of many abbreviations, and sometimes confusing terminology (e.g. movement direction vs conditions of medial and lateral movements).

2. EMG and torque was normalized to movement duration, yet movement durations were not reported. Heterogeneity in movement duration and speed of movement could significantly impact results.

3. What is the rationale for including multiple repetitions and movement directions for each muscle in the matrix for the PCA? Is the assumption that center-out and return movements are driven by the same temporal synergies?

4. It is not clear why correlations of the scores of the first PCA component between muscles can be used as a proxy of co-activation or reciprocity. Other components likely contribute to the EMG and absolute values of the scores likely matter, i.e. two muscles with only positive scores can exhibit a negative correlation but that wouldn’t mean that they are reciprocally activated. Why not simply calculate cross correlation of EMG directly?

5. It is not clear if statistics were applied correctly. E.g. an RM-ANOVA has been applied to compare the coefficient of variation between conditions. It has not been stated if and how model assumptions were tested. It is very likely that the normality of residuals has been violated given the nature of the outcome variable. Furthermore, post-hoc tests were performed when the fixed effect was not significant. The presentation and description of the statistics is also lacking. It is at times not clear what model was calculated and test statistics should be reported.

6. Equally, it is not clear if assumptions of the PCA were confirmed. E.g., are mean and covariance matrix sufficient to describe the distribution, are the intrinsic dimensions orthogonal, linearity of the low-dimensional manifold. Also, is normalization to the maximal EMG adequate? Would subtracting the mean followed by division by the standard deviation lead to different results? Peaks or baseline activity could skew the results.

7. Furthermore, it is intuitively not clear to me how temporal synergies with both positive and negative components and positive and negative scores could represent a neural signal driving multiple motor pools. How would the changes in signs be implemented on a circuit level? Wouldn’t constraining scores to be non-negative result in synergies that are easier to interpret on a circuit level?

Minor:

1. Abstract: “Results generalize …” not clear that this refers to the previous work.

2. Line 62-65: There is quite some disagreement on whether PCA is adequate to identify synergies of EMG during locomotion. And this isn’t the topic of the paper anyways. I’d remove this statement.

3. Lines 116-117: “We applied PCA with the assumption of invariant activation profiles to obtain temporal synergies” This needs more explanation.

4. Lines 185-186: Calculation of active torques should be explained in this paper.

5. LLat, RLat, etc. have not been defined.

6. Lines 200-201: Rationale for filter settings is missing. (no low pass for the raw EMG, why 10 Hz for rectified).

7. Line 244: not clear what hypothesis the authors refer to.

8. Lines 300-301: We found that intra- and inter-subject variability of EMG profiles were very low” Average standard deviation of 12% of maximal EMG does not seem that low. Also not clear if this is intra or inter subject variability.

9. Lines 350-354: Fixed effect of sex was not reported; interaction effect neither. And for some reason comparison between sexes was reported as a post-hoc test with a p value of 0.06 and discussed as not significant.

10. Lines 362-364: This conclusion is out of place and would need further elaboration.

6. PLOS authors have the option to publish the peer review history of their article (what does this mean? ). If published, this will include your full peer review and any attached files.

**Do you want your identity to be public for this peer review?** For information about this choice, including consent withdrawal, please see our Privacy Policy .

Reviewer #1: No

Reviewer #2: No

---

## [Author Response · Author response to Decision Letter 1]

14 Nov 2024

Reviewer #1:

In this paper authors explore the effect of biomechanical constraints on the dimensionality of control space. They showed that the muscle torques that support the limb against gravity are produced by more consistent combinations of muscle co-contraction than those that produce propulsion.

1. It is not clear to me why authors did PCA analysis of combined from different motor tasks data. If it were one motor task, it would show that CNS reduces control space to control this particular motor task. But what is possibly could be controlled for all different types of motions combined? Each motor task should be controlled differently. I would recommend to re-do analysis for each motor task separately.

Thank you for your comment. In this work, we have expanded upon our prior approach of analyzing reaching toward visual targets (Olesh et al. 2017 and Thomas et al. 2021). We used the PCA across movements in different directions to find the biomechanical underpinnings of muscle activation patterns which correspond to functions for supporting the limb against gravity and propulsion. Subdividing our dataset into separate movement directions and doing the PCA on each of them separately will result in the same 2 primary principal components but lower internal validity due to noise.

In response to the question of “what could be controlled for all different types of motions combined,” the simple response is force or impedance are likely the parameters controlled by CNS. The idea of the modality of control signal has been discussed in the seminal works of Bizzi, Kalaska, Geogopoulus, Kawato, Wolpert, Shadmehr, Ting, and Gribble to name a few. These and other authors have developed influential theories of internal models, which accurately capture the sensorimotor transformations that occur in the CNS with the substantial predictive power of neural signals. We have expanded our description of these ideas in the Introduction on lines 38-48, 110-138, 148-163.

PCA can be thought of as a proxy for correlation analysis. Instead of examining each correlation separately, PCA does this across all correlations. The method includes computing the covariance matrix of the data first and then reducing it to a combination of scalars (eigenvalues) and vectors (eigenvectors), the latter pointing in the directions of the largest shared variances across the whole dataset. Thus, the scores or projections of the data onto the eigenvectors can tell us if the signals are changing together (positive correlations) or reciprocally (negative correlations). The sign of the scores is the indicator of that. That is why we did regressions between signed scores (not absolute values) to investigate the co-contraction across reaching in multiple directions. We have expanded our rationale for this in Methods on lines 279-301, 337-355, 375-393. We have added a new figure 4 to illustrate these points. We have also expanded on the signed score amplitude analysis in Results, including statistical analysis, on lines 487 – 522 and show the score values in Figures 6,7,8 and Supplementary Figure S1 and S2.

Our assumption in this work is that principal components represent a low-dimensional control signal that can be produced as a result of neural activity. Some evidence supporting this assumption comes from neural recordings in non-human primates [59]. The known organization of neural circuits with multiple nested feedback loops provides clues to how the different co-contraction patterns can be produced by a reduced set of control signals. For example, the co-contraction of agonists, such as elbow flexors biceps, and brachioradialis, can be supported by common stretch reflex feedback from muscle spindles [60]. In this case, the co-contraction of agonists can be facilitated by a single descending signal that increases the gains of the stretch reflexes for these muscles. The negative reciprocal activation of antagonistic muscles, such as biceps and triceps, can be produced by inhibitory spinal interneurons, including Ia interneurons, that are known for their reciprocal inhibition action [61,62]. In this case, the reciprocal EMG can be facilitated by a single descending signal that excites the Ia interneurons for these muscles. The co-contraction of antagonistic action can be created through direct projections from the motor cortex, which is more extensive for motoneurons innervating distal muscles, through presynaptic inhibition of primary spindle afferents, the inhibition of Ia interneurons [63,64], or the spinal central pattern generator [37]. Although this analysis cannot distinguish between the contributions of different neural circuits to muscle recruitment, our method of analyzing the EMG co-contraction patterns can be used in future interventional studies, for example, with electrical or magnetic stimulation of specific neural structures, that can shed more light on the specific neural pathways that are involved in creating these patterns. We have added the descriptions of these possibilities on lines 619-639.

2. The authors wrote that there were 15 repetitions of each movement but in matrix A there is one EMG per motion. I suggest that analysis was done for each repetition separately, but it was not mentioned in the paper.

Although it is interesting to investigate the trial-to-trial variability, unfortunately, it is not possible to use our EMG data for this purpose. The EMG recordings were done using an older wired system with higher noise and motion artifacts. Only averaged EMG envelopes can be considered reliable as indicated by the EMG examples in Fig. 2C. The shaded areas show standard error, which is about 4 times smaller than the standard deviation for those traces. Furthermore, we are extending our previous conclusions from Olesh et al., 2017 to other conditions (different workspaces and to the non-dominant limb). Therefore, it is reasonable to apply the same analysis to the new data so that the results can be comparable.

3. The left part of Figure 1 is not necessary. Instead, it would be better to show locations of targets for different conditions, at least for one side.

Thank you for your suggestion, we have modified Figure 1 to show target locations for the right side.

4. Figure 2 has angles and angular velocities plotted on one panel. I would recommend using left/right different vertical axis as well using degrees instead of radians for angles.

Thank you for your suggestion, we have modified Figure 2 as requested.

5. pp.155-156 Authors wrote: “target was placed at a location that positioned the shoulder at 0 angle of all degrees of freedom and elbow at 90 degrees”. For some angles ‘0’ could be set in different positions. Was wrist pronated or supinated? Was wrist also parallel to the floor?

We have clarified the arm position description in Methods on lines 198-211. We have also included schematics of the arm position and targets in Fig. 1A to depict the DOFs at the central starting position for the lateral reaches.

6. pp.182-185 Authors wrote: “Joint angles representing 5 DOFs of the arm were derived using linear algebra, namely shoulder flexion-extension, shoulder abduction-adduction, shoulder internal-external rotation, elbow flexion-extension, forearm pronation-supination, and wrist flexion-extension”

Are there 5 or 6 DOFs?

shoulder flexion-extension(1), shoulder abduction-adduction(2), shoulder internal-external rotation(3), elbow flexion-extension(4), forearm pronation-supination(5), and wrist flexion-extension(6)

Thank you for identifying this error, we have corrected it. We incorrectly included forearm pronation-supination in the description. This degree of freedom was not included in the analysis in our earlier work. Thus, it was also not included in this analysis.

Reviewer #2:

Korol et al. examined muscle activity patterns during reaching movements in different directions and postures, performed by healthy individuals. They tested the relationship between muscle torques and muscle activity profiles, and the influence of biomechanical constraints on the dimensionality of control space. Using principal component analysis, they evaluated the contribution of individual muscles to joint torques. This work is an extension of a previous paper, the main novelty is that both dominant and non-dominant arm movements were included. They conclude that muscle torques supporting the limb against gravity are produced by more consistent combinations of muscle co-contraction than those producing propulsion. There are several issues in the manuscript that need clarification. I list them in the following.

Major:

1. The paper is quite difficult to follow. This is mainly due to the writing style, use of many abbreviations, and sometimes confusing terminology (e.g. movement direction vs conditions of medial and lateral movements).

Thank you for identifying these issues. We have replaced “movement direction” with “reaching direction” to clarify that the motion from the central to one of the outward targets constituted reaching in an outward or inward direction. We are now consistently referring to 4 conditions as workspaces for reaching by either the left or right arm. We have reduced the number of abbreviations in the Results section by spelling out the terms.

2. EMG and torque was normalized to movement duration, yet movement durations were not reported. Heterogeneity in movement duration and speed of movement could significantly impact results.

Thank you for pointing this out. We have added the mean movement durations with standard deviations across participants in the Methods section (left lateral: 0.75 +/- 0.10, left medial: 0.76 +/- 0.06, right medial: 0.74 +/- 0. 08, and right lateral: 0.71 +/- 0.08 seconds). We have reported those on lines 261-263.

3. What is the rationale for including multiple repetitions and movement directions for each muscle in the matrix for the PCA? Is the assumption that center-out and return movements are driven by the same temporal synergies?

The data was averaged across multiple repetitions. The repetitions were done to make sure we obtained reliable profiles of kinematics and EMG. We used an older wired EMG system with more noise and motion artifacts than newer systems have. We have averaged across repetitions for each reaching direction, thus the PCA matrix included only the mean EMG profiles for each reaching direction (14 forward and 14 backward) for 12 muscles. Indeed, the assumption is that the outward and inward movements are examples of reaching movements in different directions from different starting locations and thus, would be driven by the same temporal synergies that are related to the underlying forces. We have clarified this on lines 264-265 and 337-355.

4. It is not clear why correlations of the scores of the first PCA component between muscles can be used as a proxy of co-activation or reciprocity. Other components likely contribute to the EMG and absolute values of the scores likely matter, i.e. two muscles with only positive scores can exhibit a negative correlation but that wouldn’t mean that they are reciprocally activated. Why not simply calculate cross correlation of EMG directly?

PCA can be thought of as a proxy for correlation analysis. Instead of examining each correlation separately, PCA does this across all correlations. The method includes computing the covariance matrix of the data first and then reducing it to a combination of scalars (eigenvalues) and vectors (eigenvectors), the latter pointing in the directions of the largest shared variances across the whole dataset. Thus, the scores or projections of the data onto the eigenvectors can tell us if the signals are changing together (positive correlations) or reciprocally (negative correlations). The sign of the scores is the indicator of that. That is why we did regressions between signed scores (not absolute values) to investigate the co-contraction across reaching in multiple directions. We have expanded our rationale for this in Methods on lines 279-301, 337-355, 375-393. We have added a new figure 4 to illustrate these points. We have also expanded on the signed score amplitude analysis in Results, including statistical analysis, on lines 487 – 522 and show the score values in Figures 6,7,8 and Supplementary Figure S1 and S2.

The reviewer is correct in stating that the absolute values of the scores do matter as they are the magnitudes of the projection of each signal onto the eigenvector. We added a description of the implications of low PCA scores on lines 335-340 in Methods.

5. It is not clear if statistics were applied correctly. E.g. an RM-ANOVA has been applied to compare the coefficient of variation between conditions. It has not been stated if and how model assumptions were tested. It is very likely that the normality of residuals has been violated given the nature of the outcome variable. Furthermore, post-hoc tests were performed when the fixed effect was not significant. The presentation and description of the statistics is also lacking. It is at times not clear what model was calculated and test statistics should be reported.

The RM-ANOVA was applied to the coefficient of determination instead of the Pearson correlation coefficient because the former changes linearly and is normally distributed. We have added a description in Methods about verifying the ANOVA model assumptions in lines 315-323 We verified that the normality of residuals was met using Shapiro-Wilks test (custom function in MATLAB using normcdf and norminv). Due to multiple conditions included in ANOVA that can have different variances, sphericity was tested using Mauchly’s test (part of ranova function). Sphericity was violated, so the Greenhouse-Geisser correction was used in the F and p statistics reported in Results. In addition, we have tested for outliers using the z-score method (zscore function in MATLAB). Using a threshold of 3 standard deviations, no outliers were detected.

Finally, we ensured equal sample sizes for R2 across conditions.

We have expanded the description of the linear regression analysis, including correction for familywise error (alpha = 0.0008) on lines 375-393.

All F statistics, p values, and SE values from RM ANOVs are reported in the Results and Tables.

6. Equally, it is not clear if assumptions of the PCA were confirmed. E.g., are mean and covariance matrix sufficient to describe the distribution, are the intrinsic dimensions orthogonal, linearity of the low-dimensional manifold. Also, is normalization to the maximal EMG adequate? Would subtracting the mean followed by division by the standard deviation lead to different results? Peaks or baseline activity could skew the results.

We have added a description of how we addressed the PCA assumptions in Methods on lines 284-301.

We have normalized EMG to the maximal value for each muscle across reaching directions, which is a common practice in ensuring that the amplitude of the muscle activity scales within the 0 to 1 range. EMG’s interval values of the original units of millivolts are not altered by this normalization. We also mean-centered our data before using PCA to avoid the PC1 capturing the mean of the data rather than the direction of maximum variance in the temporal profiles.

We have confirmed that the covariance matrix describes the distributions of EMGs and torques sufficiently well by using Shapiro-Wilk test for normality of residuals (alpha = 0.05). We have also confirmed that the intrinsic dimensions are orthogonal by computing the dot product of each pair of eigenvectors from EMG and torques. The off-diagonal elements of the dot product matrix were approximately zero (less than 10-10), similar to an identity matrix, thus the principal components are orthogonal.

Due to the nature of EMG envelopes, it is expected that some muscles will have higher variance than others since certain reaching movements need more muscle activation than others. This meaningful heterogeneity reflects the inherent differences in the muscle activation patterns and forces they produce for different movement directions. The peaks of activity constitute meaningful variance that we want to include in our analysis. Division of the signals by the standard deviation

---

## [Decision Letter · Decision Letter 1]

28 Feb 2025

PONE-D-24-24814R1Control Signal Dimensionality Depends on Limb DynamicsPLOS ONE

Dear Dr. Gritsenko,

Thank you for submitting your manuscript to PLOS ONE. After careful consideration, we feel that it has merit but does not fully meet PLOS ONE’s publication criteria as it currently stands. Therefore, we invite you to submit a revised version of the manuscript that addresses the points raised during the review process.

Please, consider the additional analysis suggested by the reviewer 1. It will probably produce close results which could be presented in the article or in the author response.

We look forward to receiving your revised manuscript.

Kind regards,

Gennady S. Cymbalyuk, Ph.D.

Academic Editor

PLOS ONE

Journal Requirements:

Reviewers' comments:

Reviewer's Responses to Questions

**Comments to the Author**

1. If the authors have adequately addressed your comments raised in a previous round of review and you feel that this manuscript is now acceptable for publication, you may indicate that here to bypass the “Comments to the Author” section, enter your conflict of interest statement in the “Confidential to Editor” section, and submit your "Accept" recommendation.

Reviewer #1: (No Response)

Reviewer #2: All comments have been addressed

2. Is the manuscript technically sound, and do the data support the conclusions?

Reviewer #1: Partly

Reviewer #2: Yes

3. Has the statistical analysis been performed appropriately and rigorously? 

Reviewer #1: Yes

Reviewer #2: Yes

4. Have the authors made all data underlying the findings in their manuscript fully available?

Reviewer #1: Yes

Reviewer #2: Yes

5. Is the manuscript presented in an intelligible fashion and written in standard English?

Reviewer #1: Yes

Reviewer #2: Yes

6. Review Comments to the Author

Reviewer #1: A very serious issue in the methods and results of the PCA analysis remains. I agree that “The idea of the modality of control signal has been discussed in the seminal works of Bizzi, Kalaska, Geogopoulus, Kawato, Wolpert, Shadmehr, Ting, and Gribble to name a few. These and other authors have developed influential theories of internal models, which accurately capture the sensorimotor transformations that occur in the CNS with the substantial predictive power of neural signals”.

Let me clarify my concern. CNS controls one movement at time, hence suggested modules also control one movement at a time. I also support the idea that common modules may be used to control the different movements. But this happens at different times. The authors can use PCA for particular movement to extract modules for this one movement, but if authors want to extract common modules involved in controlling different movements authors need to create matrixes Anxm by combining their each movement data sets into a matrix with number of muscles (12) as one dimension and larger time dimension (time(100) x number of movements (28) )(i.e., time dimension of the set up matrix is X28 larger, while first dimension is X28 smaller) and then using PCA to find common modules.

Surfacing the assumptions is another concern. The assumption is that there are the same two primary principal components of PCA analysis of separate movement directions for both EMG and torques. Authors didn’t show any evidence that this assumption is correct. And even if it is correct this is not the correct way to find common modules, as I mentioned above.

The correct estimation of a common modules is crucial for the results and interpretation and needs to be presented.

Reviewer #2: The authors addressed all my comments and those of the other reviewer adequately. I don't have any additional comments.

7. PLOS authors have the option to publish the peer review history of their article (what does this mean? ). If published, this will include your full peer review and any attached files.

**Do you want your identity to be public for this peer review?** For information about this choice, including consent withdrawal, please see our Privacy Policy .

Reviewer #1: No

Reviewer #2: No

---

## [Author Response · Author response to Decision Letter 2]

28 Feb 2025

Reviewer #1:

Question:

A very serious issue in the methods and results of the PCA analysis remains. I agree that “The idea of the modality of control signal has been discussed in the seminal works of Bizzi, Kalaska, Geogopoulus, Kawato, Wolpert, Shadmehr, Ting, and Gribble to name a few. These and other authors have developed influential theories of internal models, which accurately capture the sensorimotor transformations that occur in the CNS with the substantial predictive power of neural signals”.

Let me clarify my concern. CNS controls one movement at time, hence suggested modules also control one movement at a time. I also support the idea that common modules may be used to control the different movements. But this happens at different times. The authors can use PCA for particular movement to extract modules for this one movement, but if authors want to extract common modules involved in controlling different movements authors need to create matrixes Anxm by combining their each movement data sets into a matrix with number of muscles (12) as one dimension and larger time dimension (time(100) x number of movements (28) )(i.e., time dimension of the set up matrix is X28 larger, while first dimension is X28 smaller) and then using PCA to find common modules.

Surfacing the assumptions is another concern. The assumption is that there are the same two primary principal components of PCA analysis of separate movement directions for both EMG and torques. Authors didn’t show any evidence that this assumption is correct. And even if it is correct this is not the correct way to find common modules, as I mentioned above.

The correct estimation of a common modules is crucial for the results and interpretation and needs to be presented.

Answer:

We appreciate the reviewer’s clarifications. The PCA method described by the reviewer is aimed at obtaining spatial synergies, while our method is aimed at obtaining temporal synergies. We have had prior communications with Andrea d’Avella who described the differences between these two methods as follows: “In most muscle synergies studies, they [muscle synergies] are defined as coordinated activations of groups of muscles, i.e. as “spatial synergies”. This definition of muscle synergy is in line with that considered by the authors, i.e. as a control signals producing force in specific directions. Indeed, scaling the time-dependent and target-dependent activation of an invariant synergy one would expect that the generated force also scales. However, while not explicitly mentioned, in this paper PCA is used to extract “temporal synergies” rather than spatial synergies. In this case, the synergies represent invariant temporal profiles that reconstruct the data through muscle-dependent and target-dependent scores.” Both types of analyses have been utilized in prior EMG studies and provide different information about the neural control mechanisms (Brambilla et al. 2023). Our introduction expands on this point as follows: “The commonly used Principal Component Analysis (PCA) can be applied across muscles, experimental conditions, and time in two ways, to obtain temporally-variable synergies (temporal synergies) or temporally invariant synergies (spatial synergies) A recent study comparing spatial and temporal synergies has shown that these two types of synergies capture different features in EMG, which potentially reflect different aspects of neural control generating them [31]. Our earlier work shows that the biomechanical dynamics during reaching is captured accurately by the temporal synergies obtained with PCA. Specifically, the temporal dynamics of muscle torques, defined as rotational forces about joints, that act to support the arm against gravity and propel it toward the goal are well described by the 1st and 2nd temporal synergies, respectively, obtained with PCA [32]. Thus, the PCA method that obtains temporal synergies ensures that the forces needed to move the limb or limb dynamics are reflected in the signals with reduced dimensionality. Arguably, these signals are more likely to be represented at the neural circuit level because of the idea of the neural embedding of musculoskeletal dynamics, or internal models mentioned above, that enables the finite CNS circuitry to efficiently control movements in multiple directions from different starting postures at different speeds [33–35]. For example, the neural circuitry of the spinal central pattern generator that controls rhythmic behaviors, such as locomotion, by producing complex EMG patterns of multiple muscles spanning multiple joints is well understood [36–40]. A recent study has shown that the temporal dynamics of this spinal central pattern generator is captured well using temporal synergies obtained with PCA of EMG during locomotion [41]. This further proves that the temporal synergies obtained with PCA can accurately capture the known neural circuitry with complex temporal dynamics.”

Here we focus on temporal synergies because we have already established the relationship between these types of synergies and muscle torques in an earlier study (Olesh et al. 2016). This answers the second part of the question above about the assumptions. Our new study aims to extend the conclusions from the earlier study that established the relationship between EMG and muscle torques using temporal synergies obtained with PCA the same way to both limbs and to other workspace locations. In the Introduction we describe how our hypothesis is based on the prior studies as flows: “The temporal profile of the envelope of surface EMG is closely related to the force the corresponding muscle is producing in response to neural activation [42]. More recently we have shown that for reaches with the dominant hand towards visual targets in three-dimensional space, the EMG profiles are closely related to the muscle torques that cause motion [32]. Specifically, the static component of EMG that underlies postural forces needed to support the arm in a specific posture and during transitions between postures is closely related to the components of muscle torques that include gravity terms in the equations of motion. This static component of EMG can be extracted as the 1st principal component of PCA applied to obtain temporal synergies [32]. This indicates that the static component of the EMG envelope from a given muscle reflects the muscle’s contribution to counteracting gravity load on the joints it spans. Moreover, the residual phasic component of EMG is closely related to the residual muscle torque that underlies the acceleration and deceleration forces toward the reaching goal after the gravity-related component is subtracted. This phasic component of EMG can be extracted as the 2nd principal component of PCA [32]. This indicates that the phasic component of the EMG envelope from a given muscle reflects the muscle’s contribution to propulsion toward the target and stopping there. Here we will test the generalizability of these relationships between muscle activity and torque profiles to the non-dominant limb and across different workspaces.”

---

## [Decision Letter · Decision Letter 2]

17 Mar 2025

Control Signal Dimensionality Depends on Limb Dynamics

PONE-D-24-24814R2

Dear Dr. Gritsenko,

We’re pleased to inform you that your manuscript has been judged scientifically suitable for publication and will be formally accepted for publication once it meets all outstanding technical requirements.

Kind regards,

Gennady S. Cymbalyuk, Ph.D.

Academic Editor

PLOS ONE

Additional Editor Comments (optional):

Reviewers' comments:

Reviewer's Responses to Questions

**Comments to the Author**

1. If the authors have adequately addressed your comments raised in a previous round of review and you feel that this manuscript is now acceptable for publication, you may indicate that here to bypass the “Comments to the Author” section, enter your conflict of interest statement in the “Confidential to Editor” section, and submit your "Accept" recommendation.

Reviewer #1: All comments have been addressed

2. Is the manuscript technically sound, and do the data support the conclusions?

Reviewer #1: Yes

3. Has the statistical analysis been performed appropriately and rigorously? 

Reviewer #1: Yes

4. Have the authors made all data underlying the findings in their manuscript fully available?

Reviewer #1: Yes

5. Is the manuscript presented in an intelligible fashion and written in standard English?

Reviewer #1: Yes

6. Review Comments to the Author

Reviewer #1: All comments were addressed, and all question were answered. However, as authors admit they did not "explicitly mentioned, in this paper PCA is used to extract “temporal synergies” rather than spatial synergies". Since most studies are about other type of synergies, I would recommend to emphasizes what type of synergies are used in this paper to avoid this confusion.

7. PLOS authors have the option to publish the peer review history of their article (what does this mean? ). If published, this will include your full peer review and any attached files.

**Do you want your identity to be public for this peer review?** For information about this choice, including consent withdrawal, please see our Privacy Policy .

Reviewer #1: No

---

## [Editor Report · Acceptance letter]

PONE-D-24-24814R2

PLOS ONE

Dear Dr. Gritsenko,

I'm pleased to inform you that your manuscript has been deemed suitable for publication in PLOS ONE. Congratulations! Your manuscript is now being handed over to our production team.

Kind regards,

on behalf of

Dr. Gennady S. Cymbalyuk

Academic Editor

PLOS ONE
